# INSTANCE DATA CONDENSATION FOR IMAGE SUPER-RESOLUTION

## ABSTRACT

Deep learning based image Super-Resolution (ISR) relies on large training datasets to optimize model generalization; this requires substantial computational and storage resources during training. While dataset condensation has shown potential in improving data efficiency and privacy for high-level computer vision tasks, it has not yet been fully exploited for ISR. In this paper, we propose a novel Instance Data Condensation (IDC) framework specifically for ISR, which achieves instance-level data condensation through Random Local Fourier Feature Extraction and Multi-level Feature Distribution Matching. This aims to optimize feature distributions at both global and local levels and obtain high-quality synthesized training content with fine detail. This framework has been utilized to condense the most commonly used training dataset for ISR, DIV2K, with a 10% condensation rate. The resulting synthetic dataset offers comparable or (in certain cases) even superior performance compared to the original full dataset and excellent training stability when used to train various popular ISR models. To the best of our knowledge, this is the first time that a condensed/synthetic dataset (with a 10% data volume) has demonstrated such performance. The associated code and synthetic dataset are available here.

## 1 INTRODUCTION

Image super-resolution (ISR) is a well-established research area in low-level computer vision, which aims to up-sample a low-resolution image to higher resolutions, while recovering fine spatial details. In recent years, deep learning inspired methods (Liang et al., 2021; Jiang et al., 2024) have been dominant in this field, offering significant improvements over conventional ISR methods based on classic signal processing theory. These learning-based solutions are typically optimized offline with a large training dataset and deployed online for processing arbitrary input images. The training data is thus crucial for maintaining the model generalization ability and for avoiding overfitting issues.

To this end, it is common practice to simply increase the amount of training material, but this introduces two primary issues. (i) Training efficiency: a large amount of training content inevitably leads to higher training costs with longer training time and greater storage/memory requirements (Paul et al., 2021). Although efforts have been made to accelerate the training process by using large batch sizes, these cannot fully reduce training costs due to increased memory consumption (Lin et al., 2022). (ii) Data quality: increased data volume does not guarantee performance improvement - large datasets can be associated with unbalanced content distributions (or bias) and data redundancy, which may result in suboptimal inference performance on certain content and reduced model generalization (Zhao et al., 2021). Moreover, privacy concerns can also arise when using large-scale data, as models can inadvertently memorize sensitive information, making them vulnerable to membership inference attacks that could potentially expose training data details (Melis et al., 2019; Lyu et al., 2020).

To address these problems, various dataset refinement approaches have been proposed, including coreset selection (Phillips, 2017; Katharopoulos & Fleuret, 2018) and dataset pruning (Ding et al., 2023; Moser et al., 2024), which result in a smaller, representative subset, derived from the large training dataset based on gradient or deep feature statistics. However, these content selection/pruning methods are constrained by the characteristics of the original content, and hence they cannot achieve optimal performance compared to the large dataset, in particular when the subset is much smaller than the original. It is also noted that dataset distillation (Wang et al., 2018a) and condensation (Zhao et al., 2021; Wang et al., 2025) techniques have been proposed recently for high-level computer vision tasks

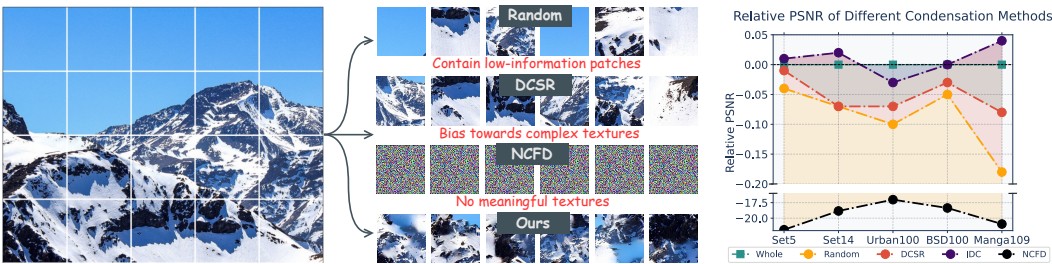

Figure 1: (**Left**): Visual comparison between synthetic patches generated by our IDC framework and those selected/synthesized by Random Selection, DCSR (Ding et al., 2023), and NCFD (v1 in our ablation study) (Wang et al., 2025). IDC's patches contain more diverse information than selection-based methods, and data condensation techniques like NCFD (designed for high-level vision) fail to produce meaningful results. (**Right**): Quantitative results show that an IDC-synthesized dataset (10% volume) can outperform the full DIV2K dataset when training ISR models.

(e.g., image classification with ground truth labels). These are designed to distill/condense a large dataset into a small but "informative" version with *synthetic* content. The aim is to achieve improved training efficiency, comparable model generalization ability (with the original dataset), and enhanced data privacy (Chen et al., 2022; Dong et al., 2022). However, these techniques cannot be directly applied to low-level tasks, such as ISR, due to the requirement for image labels. Although there have been early attempts that focus on ISR dataset condensation (Zhang et al., 2024; Dietz et al., 2025), these exhibit relatively poor performance or can only be deployed for limited application scenarios.

In this context, this paper proposes a new **Instance Data Condensation** (IDC) framework specifically for image super-resolution, which can significantly reduce the amount of training material and speed up the training process, while maintaining (or even enhancing) the model performance. Taking all the cropped patches from each individual image (instance) in a large dataset as input, this framework generates a small amount of synthetic low-resolution training patches with *condensed* information based on **Random Local Fourier Features** and **Multi-level Feature Distribution Matching**. These methods are designed to retain the distribution of the local features in the original patches and ensure the fidelity and diversity of the synthesized patches. These synthetic low-resolution patches are then up-sampled by a pre-trained ISR model to obtain their high-resolution counterparts. Targeting ISR, this framework has been utilized to condense the most commonly used training dataset in the ISR literature, DIV2K (Agustsson & Timofte, 2017), resulting in a synthetic dataset with only 10% training patches. The condensed dataset was then used to train three popular ISR network architectures, EDSR (Lim et al., 2017), SwinIR (Liang et al., 2021) and MambaIRv2 (Guo et al., 2024a), which achieve similar or even better evaluation performance compared to the same networks trained with the full DIV2K dataset. The main contributions of this work are summarized below.

1. We propose a new data condensation framework specifically for ISR that operates at the **instance (image) level**, a paradigm that effectively bypasses the need for class labels common in high-level vision tasks. It is the **first time** that a highly condensed dataset has achieved better performance than the original full dataset when used for training ISR models (as shown in Figure 1.(**Right**)).

2. We design a **Multi-level Feature Distribution Matching** approach, which learns feature distributions at instance and group levels. This hierarchical strategy progressively refines the synthetic data, enhancing feature quality and diversity and leading to well-conditioned samples, enabling high-quality visual feature learning in distribution matching, as shown in Figure 1.(**Left**).

3. We develop the novel **Random Local Fourier Features**, which captures high-frequency details and local features to facilitate distribution matching in learning high-fidelity synthetic images.

## 2 RELATED WORK

**Image Super-Resolution (ISR)** is a fundamental task in low-level computer vision that aims to reconstruct high-resolution (HR) images from their low-resolution (LR) counterparts. Over the past decade, advances in deep learning have significantly improved ISR performance, employing ISR models based on network architectures including convolutional neural networks (CNNs) (Kim et al.,

2016; Zhang et al., 2018), vision transformers (Wang et al., 2022c; Jiang et al., 2025), and structured state-space models (Guo et al., 2024b; Shi et al., 2025). These ISR methods, due to their data-driven nature, are typically trained offline on a large training dataset and then deployed online for real-world applications. In this case, the training dataset is critical to model performance and generalization.

To facilitate ISR model training, a series of datasets have been developed including small ones such as T91 (Yang et al., 2010) and BSD200 (Martin et al., 2001), which contain 91 and 200 natural images, respectively. A significant milestone was the release of DIV2K (Agustsson & Timofte, 2017), consisting of 800 high-resolution images, which is now the most commonly used training dataset for ISR. Other alternatives also exist including Flickr2K (Timofte et al., 2017) that comprises 2,650 high-resolution images collected from online sources, and LSDIR (Li et al., 2023) with more than 85,000 images. In this paper, we employ DIV2K as the original training dataset to demonstrate the data condensation process due to its popularity. Based on the common practice in ISR literature (Wang et al., 2022b), existing methods typically crop each image in the training dataset into small patches. For example, the 800 original images in DIV2K (Agustsson & Timofte, 2017) along with their corresponding low-resolution counterparts (e.g., down-sampled by a factor of 4) are partitioned into overlapped patches, producing approximately 120K pairs (LR and HR), which are used in the training process as the input (LR) and output target (HR) of ISR models.

**Dataset condensation and distillation** aims to condense or distill a large-scale original training set into a smaller synthetic one, capable of offering comparable performance on unseen test data when used to train a model for a downstream task. Based on the optimization objectives, current dataset condensation methods can be categorized into three classes: performance matching, parameter matching, and distribution matching. *Performance matching* based approaches (Wang et al., 2018a; Zhou et al., 2022) are designed to optimize the synthetic dataset to achieve minimum training loss (for the downstream task) compared to using the original training dataset, while *parameter matching* (Yu et al., 2024) encourages models trained on the synthetic dataset to maintain consistency in the parameter space with those trained on the original dataset. Both types of approaches are similar to bi-level meta-learning (Finn et al., 2017) and gradient-based hyperparameter optimization techniques (Du et al., 2023), which are associated with high computation graph storage costs and are difficult to apply to high resolution and large scale datasets (Cazenavette et al., 2022; Yu et al., 2023). *Distribution matching* generates synthetic data by directly optimizing the distribution distance between synthetic and real data, which typically leverages feature embeddings extracted from networks with different initializations to construct the distribution space, utilizing Maximum Mean Discrepancy as the distribution metric (Zhao et al., 2023). This has been further enhanced by incorporating batch normalization statistics (Yin et al., 2023; Du et al., 2024; Shao et al., 2024) or neural features in the complex plane (Wang et al., 2025), to achieve enhanced performance on large-scale datasets.

For ISR, multiple contributions have targeted the generation of a small subset of representative samples, based on diversity criteria such as texture complexity and blockiness distributions (Ding et al., 2023; Ohtani et al., 2024). Given the distinct differences between image classification and ISR, it is difficult to directly apply the dataset distillation/condensation techniques mentioned above to the ISR task (Liu et al., 2021). To the best of our knowledge, there has been very limited investigation into this research topic, with only one notable work (Zhang et al., 2024), which utilizes GAN-based pretrained models to generate synthetic training images. However, this is still only applicable on an SR dataset with labels (Wang et al., 2018b).

## 3 METHOD

**Data condensation for ISR.** The training of ISR models employs paired high- and low-resolution (HR and LR) image patches, typically with a spatial resolution larger than $192 \times 192$ (for HR patches) (Lim et al., 2017; Liang et al., 2021; Guo et al., 2024a). However, most current mainstream dataset condensation methods proposed for high-level vision tasks are designed and validated on much lower resolution content (e.g. $32 \times 32$ or $64 \times 64$) (Wang et al., 2018a; Zhao et al., 2021; Wang et al., 2025). As a result, there are significant challenges when these methods are directly applied to the ISR task. First, the training with high-resolution images that contain more pixels requires significantly increased optimization space (e.g., features, gradients, etc.), resulting in a much slower (or even impossible) training process. This is in particular relevant to the dataset condensation methods based on performance or gradient matching. Secondly, existing data condensation methods often rely

Figure 2: Illustration of the proposed Instance Data Condensation (IDC) framework.

on class labels to calculate task losses (such as the cross-entropy loss, soft labels, etc.) to guide the optimization of synthetic samples, while commonly used datasets in ISR tasks (e.g., DIV2K (Agustsson & Timofte, 2017)) are not associated with real class labels.

In this paper, our approach is based on distribution matching, which minimizes the distance of distributions between two datasets. Specifically, given a real, original dataset $\mathcal{T}$ to a synthetic dataset $\mathcal{S}$, $|\mathcal{S}| \ll |\mathcal{T}|$. The optimization goal is formulated as:

$$\mathcal{S} = \underset{\mathcal{S}}{\arg\min} \, \mathcal{L}(\mathcal{S}, \mathcal{T}), \tag{1}$$

where $\mathcal{L}$ stands for the distance measurement function.

**Distance measurement functions.** Early approaches (Wang et al., 2022a; Zhao & Bilen, 2023) employ Mean Squared Error or Maximum Mean Discrepancy as $\mathcal{L}$, while recently, Neural Characteristic Function Discrepancy (NCFD) (Wang et al., 2025) has been proposed to capture distributional discrepancies by aligning the phases and amplitudes of neural features in the complex plane, achieving a balance between realism and diversity in the synthetic samples. Specifically, the optimization process is described by:

$$\min_{D_\mathcal{S}} \max_{\psi} \mathcal{L}_{dist}(D_\mathcal{T}, D_\mathcal{S}, f, \psi) = \min_{D_\mathcal{S}} \max_{\psi} \mathbb{E}_{\boldsymbol{x} \sim D_\mathcal{T}, \hat{\boldsymbol{x}} \sim D_\mathcal{S}} \int_t \sqrt{\mathrm{Chf}(\boldsymbol{t}; f)} \, dF(\boldsymbol{t}; \psi), \tag{2}$$

$$\mathrm{Chf}(\boldsymbol{t}; f) = \quad \alpha \underbrace{\left( \left( \left| \Phi_{f(\boldsymbol{x})}(\boldsymbol{t}) - \Phi_{f(\hat{\boldsymbol{x}})}(\boldsymbol{t}) \right| \right)^2 \right)}_{\textit{amplitude difference}}$$
$$+ \underbrace{(1 - \alpha) \cdot (2 \left| \Phi_{f(\boldsymbol{x})}(\boldsymbol{t}) \right| \left| \Phi_{f(\hat{\boldsymbol{x}})}(\boldsymbol{t}) \right|) \cdot (1 - \cos(\boldsymbol{a}_{f(\boldsymbol{x})}(\boldsymbol{t}) - \boldsymbol{a}_{f(\hat{\boldsymbol{x}})}(\boldsymbol{t})))}_{\textit{phase difference}}. \tag{3}$$

Here $D$ denotes the distribution of a dataset. $f$ is the feature extractor that maps input training samples, i.e., $\boldsymbol{x} \in \mathcal{T}$ or $\hat{\boldsymbol{x}} \in \mathcal{S}$, into the latent space. $F(\boldsymbol{t}, \psi)$ is the cumulative distribution function of the frequency argument $\boldsymbol{t}$, while $\psi$ is a parameterized sampling network to obtain the distribution of $\boldsymbol{t}$. $\Phi_{f(\boldsymbol{x})}(\boldsymbol{t}) = \mathbb{E}_{f(\boldsymbol{x})} \left[ e^{j\langle \boldsymbol{t}, f(\boldsymbol{x}) \rangle} \right]$ stands for the characteristic function. $\mathrm{Chf}(\boldsymbol{t}; f)$ calculates the distributional discrepancy between training and synthesized samples in the complex plane - here *phase* stands for data centers which is crucial for realism, and *amplitude* captures the distribution scale, contributing to the diversity. $\alpha$ is a hyper-parameter to balance the amplitude and phase information during the optimization process.

It is noted that while NCFD (Wang et al., 2025) has contributed to the SOTA dataset condensation method for image classification, it cannot capture sufficient high-frequency details and rich local semantic information when directly applied to condense training datasets for ISR. This is illustrated in Figure 1. This may be because the high-dimensional feature distribution is intractable, directly learning the joint distribution of features with dimensions $C \times H \times W$. Furthermore, it uses a random Gaussian matrix for mapping features - $f(\cdot)^{N \times C \times H \times W} \mapsto f(\cdot)^{N \times D}$ (when the sampling network is not used (Wang et al., 2025)) - (i) the random Gaussian projection fuses information globally, which, however, is unsuitable for many low-level vision tasks including ISR, where the fine-grained local information; (ii) it does not capture the high-frequency features within high-resolution feature maps extracted by ISR models, thus limiting the capacity to learn fine details in the synthesis process.

### 3.1 INSTANCE DATA CONDENSATION

To address the above issues, specifically for the ISR task, we propose a novel **Instance Data Condensation** (IDC) framework, which performs distribution matching for the local features at multiple levels. This approach efficiently handles the requirements of high-resolution patches for training ISR models and effectively preserves high-frequency details in the original dataset. The proposed IDC framework, shown in Figure 2, generates synthetic training patch pairs in two stages.

In the first stage, given a real training dataset $\mathcal{T}$, which contains training patch pairs cropped from $C$ original, high-resolution (HR) images and their low-resolution counterparts, we first perform teacher ISR model training, which will be used in the distribution matching process. We then **consider each image as an individual class**, and take its corresponding, LR training patches $\boldsymbol{x} \in \mathbb{R}^{N \times 3 \times H \times W}$ as the input of the distribution matching process. A set of synthetic patches $\hat{\boldsymbol{x}} \in \mathbb{R}^{n \times 3 \times H \times W}$ will then be registered as learnable parameters, which are eventually the LR output synthetic patches of the IDC framework. To reduce computational cost, we utilize the feature extractor $f$ in the teacher ISR model to map $\boldsymbol{x}$ and $\hat{\boldsymbol{x}}$ into the latent space, resulting in $f(\boldsymbol{x})$ and $f(\hat{\boldsymbol{x}})$, similar to previous works in distribution matching (Deng et al., 2024; Wang et al., 2025). Here $N$ is the total number of patches cropped from the current RGB image/class. $H \times W$ represents the spatial resolution of the training patches. $r \in (0, 1)$ is the condensation ratio, i.e. $n = N * r$.

As mentioned above, the latest distribution matching method (Wang et al., 2025) employs a random Gaussian matrix for global matching, which works well when representing the feature distributions of image classification task, but cannot capture local high-frequency details that are essential for ISR. In this work, we propose to use **Random Local Fourier Features**, which can represent more fine-grained local features, and unfold the obtained feature maps into a batch of features for performing distribution matching. This enables the capturing of fine local details. As shown in Figure 2, both the real and synthetic feature maps, $f(\boldsymbol{x})$ and $f(\hat{\boldsymbol{x}})$, are processed to obtain their corresponding local features, $f'(\boldsymbol{x})$ and $f'(\hat{\boldsymbol{x}})$. Given the extracted local features, we then optimize a **multi-level feature distribution loss**, consisting of instance-level distribution ($\mathcal{L}_{ins}$), group-level distribution ($\mathcal{L}_{group}$), and pair-wise ($\mathcal{L}_{pair}$) losses, in order to obtain the synthetic LR patches $\hat{\boldsymbol{x}}$ for this image/class.

In the second stage, we employ the **pre-trained teacher ISR model to up-sample** $\hat{\boldsymbol{x}}$ to obtain their corresponding HR patches $\hat{\boldsymbol{x}}^{HR}$, which effectively leverages knowledge distillation (Hinton et al., 2015) by directly exploiting the LR-to-HR mapping of the teacher model.

### 3.2 RANDOM LOCAL FOURIER FEATURES

Given that the extracted feature maps of the real training patches are denoted as $f(\boldsymbol{x}) \in \mathbb{R}^{N \times c \times h \times w}$, in which $c, h, w$ are the channel number, size of the feature map, respectively, we first define an identity matrix $I_{C' \times C'}$ and reshape it into a convolutional filter, $E \in \mathbb{R}^{C' \times c \times k \times k}$ where $C' = c \times k^2$, and $k$ is the kernel size. This filter is used to map $f(\boldsymbol{x})$ from the channel-spatial domain to the channel domain to extract local features, while keeping the spatial structure information. To extract the high frequency details, we further apply the Fourier transform $\mathcal{F}$ after extracting local features by $E$, which in practice, is achieved by directly applying $\mathcal{F}$ to the convolutional filter $E$ in the output channel dimension, and decomposing the real and imaginary parts to form a new Fourier-based convolution filter $F \in \mathbb{R}^{2C' \times c \times k \times k}$:

$$F = [\Re(\mathcal{F}(E)), -\Im(\mathcal{F}(E))]. \tag{4}$$

The feature maps, $f(\boldsymbol{x})$, are transformed by $F$, to a more informative representation, $f'(\boldsymbol{x})$, which fully encodes the local spatial and channel information in the frequency domain that is particularly suitable for data and textures with periodic structures.

$$f'(\boldsymbol{x}) = f(\boldsymbol{x}) \circledast F. \tag{5}$$

Here, to reduce the size of $f'(\boldsymbol{x})$ and the complexity for loss computation, we implement channel-wise random sampling to get $F \in \mathbb{R}^{C_{out} \times c \times k \times k}, C_{out} < C'$ (Wang et al., 2025). We also apply batch normalization to $f(\boldsymbol{x})$ before convolution.

Finally, we partition $f'(\boldsymbol{x})$ to obtain local feature patches $f''(\boldsymbol{x}) \in \mathbb{R}^{N' \times C_{out} \times p \times p}$, where $N' = N * \lceil h/p \rceil * \lceil w/p \rceil$, and $p$ is the patch size. In order to learn local features in the later multi-level distribution matching, we further *unfold* $f''(\boldsymbol{x})$ from $\mathbb{R}^{N' \times C_{out} \times p \times p}$ to $\mathbb{R}^{(N' \times p \times p) \times C_{out}}$, which treats features in the patches as different samples in a batch.

This operation (Random Local Fourier Features) has also been applied to the feature maps of the synthetic patches, $f(\hat{\boldsymbol{x}})$, before performing the multi-level distribution matching detailed below.

### 3.3 MULTI-LEVEL DISTRIBUTION MATCHING

To adapt to the nature of the ISR task, rather than directly using existing distribution matching losses, we propose a Multi-level Distribution Matching approach to optimize the feature distribution of synthetic patches at both instance (class) and group levels and match pair-wise feature patches.

**Matching instance-level feature distributions.** After obtaining local features for both real and synthetic patches, $f''(\boldsymbol{x})$ and $f''(\hat{\boldsymbol{x}})$, the instance-level feature distribution loss, $\mathcal{L}_{ins}$, is calculated to minimize the distributional discrepancy between the real and synthetic patches. Here we do not use the sampling network mentioned in its original literature (Wang et al., 2025) for efficient computation:

$$\mathcal{L}_{ins} = \mathcal{L}_{dist}(\boldsymbol{x}, \hat{\boldsymbol{x}}, f) = \mathbb{E}_{\boldsymbol{x}, \hat{\boldsymbol{x}}} \int_{\boldsymbol{t}} \sqrt{\mathrm{Chf}(\boldsymbol{t}; f)} \, dF(\boldsymbol{t}). \tag{6}$$

**Matching group-wise feature distributions.** While instance-level feature distribution matching ensures consistency in the overall (global) feature distribution between synthetic and original patches, local features often exhibit significant complexity and diversity, preventing instance-level matching from sufficiently capturing all distributional differences. To address this issue, we designed a more fine-grained, group-level feature distribution matching loss, $\mathcal{L}_{group}$, which first partitions the real local features $f''(\boldsymbol{x})$ into $M$ groups using K-means clustering. Each synthetic local feature is then iteratively assigned to its nearest group centroid with a progressive assigning strategy.

Here, the number of synthetic features assigned to each group is proportional to the number of real features in that group, and the assignment is updated in steps to maintain stable optimization. To ensure that the assignments of neighboring features are consistent and to reduce the complexity, the unfolded local features which originally come from the same local feature patch, are grouped together. The resulting grouped local features are denoted as $\{g_m(\boldsymbol{x})\}$ and $\{g_m(\hat{\boldsymbol{x}})\}$, $m = 1, \ldots, M$.

After this assignment, we compute the feature distribution matching loss for each group to more accurately learn the synthetic data, by reflecting the real data distribution at the group level. This is described by the following equation:

$$\mathcal{L}_{group} = \sum_{m=1}^{M} \mathcal{L}_{dist}(g_m(\boldsymbol{x}), g_m(\hat{\boldsymbol{x}}), id(\cdot)), \tag{7}$$

in which $id$ denotes the identity function, i.e. $id(x) = x$.

**Matching pair-wise features.** Moreover, to further address the unique nature of the ISR task, which aims to reconstruct content with high fidelity and fine local details, we also introduce a pair-wise loss, $\mathcal{L}_{pair}$, within each local feature group described above. Specifically, for each synthetic feature patch assigned to a group, we identify the most similar real local feature patch within the same group and construct a pair. We then compute the L1 loss between the paired features to minimize their discrepancy. This is expected to encourage the synthetic data to better match the real data at the local detail level and to improve the fidelity of fine details and the overall visual quality of the synthesized images. This process can be described by the equation below:

$$\mathcal{L}_{pair} = \sum_{m=1}^{M} \sum_{i=1}^{N_m} \frac{1}{N_m} \| f''(\boldsymbol{x})^{g_m(i)} - f''(\hat{\boldsymbol{x}})^{g_m(i)} \|_1, \tag{8}$$

in which $N_m$ is the number of feature patch pairs in the group $m$. $f''(\boldsymbol{x})^{g_m(i)}$ represents the real feature in the $i^{\text{th}}$ pair, and $f''(\hat{\boldsymbol{x}})^{g_m(i)}$ is the corresponding synthetic feature in this pair.

These three losses are used at different stages in the proposed IDC framework for synthetic data optimization, and the training algorithm is described in Algorithm 1.

## 4 EXPERIMENT CONFIGURATION

**Implementation details.** IDC is a novel dataset condensation method specifically designed for the ISR task, which can be applied to any dataset with or without labels. In this experiment, we employ

---

**Algorithm 1:** Instance Data Condensation.

---

**Input:** Training dataset $\mathcal{T} = \{[\boldsymbol{x}_c, \boldsymbol{x}_c^{HR}], c = 1, \dots, C\}$; Synthetic dataset $\mathcal{S} = \varnothing$; ISR model $f$
   pre-trained on $\mathcal{T}$; image/class index $c$; condense ratio $r$; instance-level distribution loss weight $w_{ins}$;
   group-level distribution loss weight $w_{group}$; pair-wise loss weight $w_{pair}$; learning rate $\eta$.
**for** *each real patch $\boldsymbol{x}_c$ in $\mathcal{T}$* **do**
    Randomly initialize the synthetic patches $\hat{\boldsymbol{x}}_c$, subject to $|\hat{\boldsymbol{x}}_c| = r|\boldsymbol{x}_c|$;
    Extract the feature maps using $f$ for $\boldsymbol{x}_c$;
    **for** *i in num_iters* **do**
        Extract the feature maps using $f$ for $\hat{\boldsymbol{x}}_c$;          ▷ Subsection 3.2
        Extract their Random Local Fourier features and Unfold the feature maps into local patches;
        **if** *i in warm up* **then**
            Compute $\mathcal{L} = w_{ins}\mathcal{L}_{ins}$;
        **else**                                    ▷ Subsection 3.3
            **if** *i in assigning* **then**
                Increase *grouping*;
                Increase *pairing*;
            Compute $\mathcal{L} = w_{ins}\mathcal{L}_{ins} + w_{group}\mathcal{L}_{group} + w_{pair}\mathcal{L}_{pair}$ ;
        Update $\hat{\boldsymbol{x}}_c \leftarrow \hat{\boldsymbol{x}}_c - \eta\nabla_{\hat{\boldsymbol{x}}_c}\mathcal{L}$;
    $\hat{\boldsymbol{x}}_c^{HR} = f(\hat{\boldsymbol{x}}_c) \uparrow$;
    $\mathcal{S} = \mathcal{S} \cup \{[\hat{\boldsymbol{x}}_c, \hat{\boldsymbol{x}}_c^{HR}]\}$;
**Output:** $\mathcal{S}$

---

SwinIR (Liang et al., 2021) and MambaIRv2 (Guo et al., 2024a) that are optimized on the original real training dataset $\mathcal{T}$ obtained from DIV2K as the feature extractor and the up-sampling ISR model, respectively. For each instance/class, we can generate the synthetic dataset with a 10% condense ratio for 20k iterations with a single Nvidia-A40 GPU. For HR patch up-sampling, we choose one of the latest ISR models, MambaIRv2. Here, the outputs of up-sampling ISR models (teacher) act as a form of knowledge distillation, providing regularized targets that guide the model to learn more generalizable features (The effect of different up-sampling ISR models is provided in Appendix D). Other training and hyper-parameter configurations are also provided in Subsection C.1.

**Datasets and metrics.** To evaluate the performance of the proposed method, we used the widely used training dataset for ISR, DIV2K (Agustsson & Timofte, 2017), as the original training dataset; this contains 800 images with a 2K resolution. We follow the common practice (Wang et al., 2022b; Lim et al., 2017; Zhang et al., 2018; Wang et al., 2022c; Liang et al., 2021; Shi et al., 2025) to crop all the images into overlapped patches with a spatial resolution of $256{\times}256$ and down-sample them into corresponding LR patches ($64{\times}64$) targeting the $\times 4$ task. This results in a total number of 120,765 HR-LR patch pairs, forming the real dataset $\mathcal{T}$ mentioned above. For performance evaluation, we use five commonly used datasets (Bevilacqua et al., 2012; Zeyde et al., 2012; Huang et al., 2015; Martin et al., 2001; Fujimoto et al., 2016) and perform the $\times 4$ ISR task. The performance has been measured by PSNR and SSIM (Wang et al., 2004).

**Benchmarks.** We compared the proposed methods with four dataset core-selection/pruning methods, including random selection, Herding (Welling, 2009), Kcenter (Sener & Savarese, 2018) and DCSR (Ding et al., 2023). All these benchmarks were applied on $\mathcal{T}$ with the same ratio (10%) to obtain the selected/pruned (real) patches and the implementation details are provided in Subsection C.2. We also provide the results based on the whole original training set $\mathcal{T}$ for reference. We did not compare IDC to the only available dataset condensation method designed for ISR, GSDD (Zhang et al., 2024), because it is only suitable for datasets with classification labels and was evaluated with GAN-based ISR models, which is not the common practice in ISR literature. Moreover, we did not directly benchmark our approach against dataset condensation methods proposed for high-level vision tasks due to their label-based nature. However, we do take them into account in the ablation study below.

**ISR models.** Three popular ISR models including EDSR-baseline (Lim et al., 2017), SwinIR (Liang et al., 2021) and MambaIRv2 (Guo et al., 2024a) have been trained here on different datasets generated by the proposed and benchmark methods in this experiment based on the same training configurations. The training and evaluation configurations for ISR methods are summarized in Subsection C.3.

Table 1: Comparison with coreset selection and dataset pruning methods. For all methods (except the Whole), we generated/selected 12,076 LR-HR pairs with the condense ratio $r$=10%. The results are based on PSNR (dB) and SSIM. The best and second best results are highlighted in red and yellow cells, respectively.

| PSNR (dB)↑ | Set5 (Bevilacqua et al., 2012) | | | Set14 (Zeyde et al., 2012) | | | Urban100 (Huang et al., 2015) | | |
|---|---|---|---|---|---|---|---|---|---|
| | EDSR | SwinIR | MambaIRv2 | EDSR | SwinIR | MambaIRv2 | EDSR | SwinIR | MambaIRv2 |
| Whole | 30.17 | 30.28 | 30.52 | 26.61 | 26.78 | 26.88 | 24.50 | 24.92 | 25.21 |
| Random | 30.13 | 30.20 | 30.49 | 26.54 | 26.68 | 26.86 | 24.40 | 24.70 | 25.02 |
| Herding (Welling, 2009) | 29.94 | 30.03 | 30.36 | 26.43 | 26.54 | 26.65 | 24.10 | 24.40 | 24.65 |
| Kcenter (Sener & Savarese, 2018) | 30.01 | 30.12 | 30.45 | 26.50 | 26.61 | 26.79 | 24.28 | 24.56 | 24.86 |
| DCSR (Ding et al., 2023) | 30.16 | 30.33 | 30.51 | 26.54 | 26.69 | 26.86 | 24.43 | 24.78 | 25.10 |
| **IDC (ours)** | 30.18 | 30.34 | 30.52 | 26.63 | 26.78 | 26.91 | 24.47 | 24.86 | 25.16 |

| PSNR (dB)↑ | BSD100 (Martin et al., 2001) | | | Manga109 (Fujimoto et al., 2016) | | |
|---|---|---|---|---|---|---|
| | EDSR | SwinIR | MambaIRv2 | EDSR | SwinIR | MambaIRv2 |
| Whole | 26.24 | 26.34 | 26.40 | 28.50 | 28.98 | 29.22 |
| Random | 26.19 | 26.28 | 26.37 | 28.32 | 28.74 | 29.07 |
| Herding (Welling, 2009) | 26.11 | 26.19 | 26.30 | 27.88 | 28.30 | 28.54 |
| Kcenter (Sener & Savarese, 2018) | 26.17 | 26.25 | 26.34 | 28.19 | 28.64 | 28.89 |
| DCSR (Ding et al., 2023) | 26.21 | 26.31 | 26.39 | 28.42 | 28.86 | 29.14 |
| **IDC (ours)** | 26.24 | 26.34 | 26.40 | 28.54 | 29.02 | 29.26 |

| SSIM↑ | Set5 (Bevilacqua et al., 2012) | | | Set14 (Zeyde et al., 2012) | | | Urban100 (Huang et al., 2015) | | |
|---|---|---|---|---|---|---|---|---|---|
| | EDSR | SwinIR | MambaIRv2 | EDSR | SwinIR | MambaIRv2 | EDSR | SwinIR | MambaIRv2 |
| Whole | 0.8645 | 0.8679 | 0.8699 | 0.7435 | 0.7492 | 0.7507 | 0.7627 | 0.7769 | 0.7861 |
| Random | 0.8639 | 0.8664 | 0.8694 | 0.7424 | 0.7469 | 0.7498 | 0.7594 | 0.7707 | 0.7801 |
| Herding (Welling, 2009) | 0.8607 | 0.8646 | 0.8674 | 0.7397 | 0.7439 | 0.7455 | 0.7504 | 0.7617 | 0.7701 |
| Kcenter (Sener & Savarese, 2018) | 0.8626 | 0.8655 | 0.8686 | 0.7412 | 0.7452 | 0.7481 | 0.7558 | 0.7674 | 0.7759 |
| DCSR (Ding et al., 2023) | 0.8642 | 0.8679 | 0.8696 | 0.7426 | 0.7478 | 0.7504 | 0.7608 | 0.7737 | 0.7825 |
| **IDC (ours)** | 0.8644 | 0.8680 | 0.8702 | 0.7445 | 0.7487 | 0.7516 | 0.7610 | 0.7749 | 0.7824 |

| SSIM↑ | BSD100 (Martin et al., 2001) | | | Manga109 (Fujimoto et al., 2016) | | |
|---|---|---|---|---|---|---|
| | EDSR | SwinIR | MambaIRv2 | EDSR | SwinIR | MambaIRv2 |
| Whole | 0.7137 | 0.7185 | 0.7202 | 0.8788 | 0.8877 | 0.8903 |
| Random | 0.7126 | 0.7172 | 0.7195 | 0.8767 | 0.8854 | 0.8889 |
| Herding (Welling, 2009) | 0.7102 | 0.7145 | 0.7174 | 0.8704 | 0.8792 | 0.8822 |
| Kcenter (Sener & Savarese, 2018) | 0.7130 | 0.7162 | 0.7185 | 0.8776 | 0.8839 | 0.8860 |
| DCSR (Ding et al., 2023) | 0.7125 | 0.7178 | 0.7202 | 0.8773 | 0.8866 | 0.8895 |
| **IDC (ours)** | 0.7137 | 0.7184 | 0.7198 | 0.8795 | 0.8882 | 0.8895 |

## 5 RESULTS AND DISCUSSION

**Overall performance.** Table 1 summarizes the quantitative results of our proposed IDC approach and other dataset selection/pruning methods for ISR, while the comparison of reconstruction quality for different ISR methods is provided in Appendix F. It can be observed that IDC consistently achieves superior performance compared to the benchmark methods, across most of the test datasets and quality metrics. In particular, with only 10% of the data volume, it offers even better evaluation performance compared to the whole original training set on four out of five datasets. To further validate the robustness of our framework, we conducted experiments with a more aggressive 1% condensation ratio and evaluated our framework on other datasets, with detailed results presented in Appendix E. Moreover, we provide visual examples of synthetic training patches in Figure 3, which show that IDC can preserve high-frequency details and texture information. As far as we are aware, this is the first data condensation method that achieves this level of performance for the ISR task.

Alongside evaluation performance, training datasets should also support reduced optimization time and stable training behavior. To assess these characteristics, we plot the performance comparison between our proposed IDC method, Random selection, and the DCSR approach on the Set14 dataset with different learning rates during the training process, using SwinIR as the ISR model with a 1% condensation ratio (a more challenging case). It has been observed that the DCSR method, which employs Sobel filters to preserve real high-gradient image regions, exhibits evident overfitting behavior across all four different learning rate configurations. In contrast, our method demonstrates better stability and generalization capability, maintaining a steady upward trend throughout the training process. More analysis and limitations are discussed in the Appendix G.

**Ablation study.** In order to compare our approach with SOTA dataset condensation methods proposed for high-level vision tasks, we first start with the original NCFD (Wang et al., 2025), and progressively add each component, resulting in four variants v1-v4. In addition, to thoroughly verify the contributions of the main components in the proposed IDC framework, we further obtained another three variants v5-v7 by removing Unfolding, Local Feature, Instance Loss, Group/Pair Losses

Table 2: Results of the ablation study. ✓: included, ✗: excluded. Here, we only condense 80 images/classes for each setting but with the same condense ratio 10% in each class (effectively 1% condense ratio for the whole dataset). We choose the EDSR-baseline (Lim et al., 2017) as the ISR model, evaluated on three test datasets.

| Variant | Local Feature | Unfolding | Instance Loss | Group Loss | Pair Loss | Set5 | | Urban100 | | Manga109 | |
|---------|---------------|-----------|---------------|------------|-----------|------|------|----------|------|----------|------|
| | | | | | | PSNR | SSIM | PSNR | SSIM | PSNR | SSIM |
| **IDC** | ✓ | ✓ | ✓ | ✓ | ✓ | 30.02 | 0.8616 | 24.07 | 0.7474 | 28.00 | 0.8723 |
| v1 | ✗ | ✗ | ✓ | ✗ | ✗ | -21.71 | -0.7606 | -16.60 | -0.6674 | -20.44 | -0.7499 |
| v2 | ✓ | ✗ | ✓ | ✗ | ✗ | -0.49 | -0.0092 | -0.60 | -0.0247 | -0.56 | -0.0108 |
| v3 | ✓ | ✓ | ✓ | ✗ | ✗ | -0.16 | -0.0024 | -0.12 | -0.0059 | -0.04 | -0.0015 |
| v4 | ✓ | ✓ | ✓ | ✓ | ✗ | -0.05 | -0.0008 | -0.04 | -0.0020 | 0.06 | 0.0007 |
| v5 | ✗ | ✓ | ✓ | ✓ | ✓ | -0.19 | -0.0023 | -0.04 | -0.0017 | -0.06 | -0.0005 |
| v6 | ✓ | ✗ | ✓ | ✓ | ✓ | -0.29 | -0.0034 | -0.10 | -0.0048 | -0.10 | -0.0017 |
| v7 | ✓ | ✓ | ✗ | ✓ | ✓ | -0.22 | -0.0032 | -0.02 | -0.0012 | -0.09 | -0.0011 |

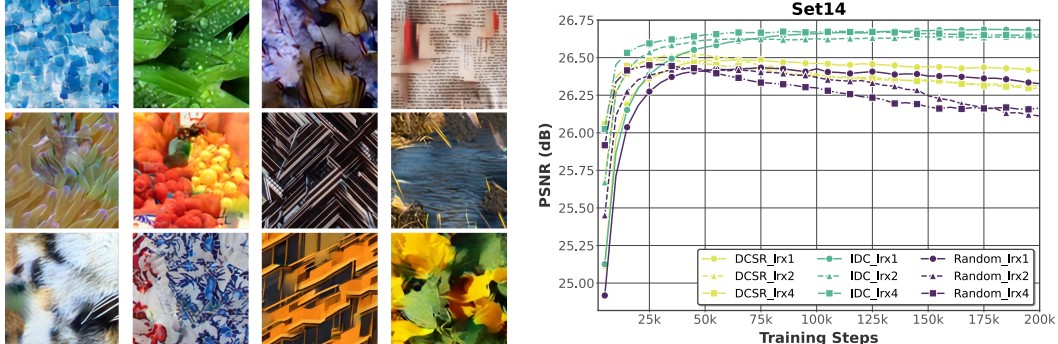

Figure 3: (**Left**): Visual Examples of our synthetic images. (**Right**): Validation trajectory on the Set14.

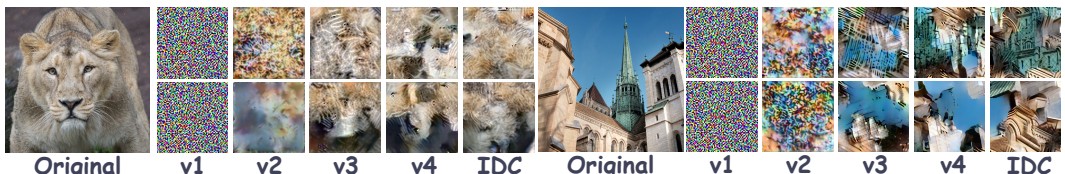

Figure 4: Starting from the original NCFD (v1), the visual evolution for adding each contribution.

(v3 already exists)[1], or Pair Loss (v4 already exists), respectively. The ablation results, based on Set5, Urban100 and Manga109 datasets, are shown in Table 2, which confirms the effectiveness and necessity of each component in the IDC framework. More experiment and analysis for the novel RLFF are shown in Subsection D.3. We also provide their synthetic patch examples of v1-v4 in Figure 4 for visualization. More visual examples are provided in the Appendix F.

## 6 CONCLUSION

This paper presents Instance Data Condensation (IDC) specifically targeting image super-resolution. It synthesizes a small yet informative training dataset from a large dataset containing real images. By leveraging a multi-level distribution matching framework and the new Random Local Fourier Features, IDC captures essential structural and textural features from the original high-resolution images and achieves significant data condensation. Trained on the resulting small synthetic dataset (with only 10% of the original data volume) ISR models can achieve comparable or (in some cases) even superior performance compared to using the entire real training dataset (DIV2K), when benchmarked on multiple test datasets. As far as we are aware, this is the first data condensation approach designed specifically for ISR capable of this level of performance. More importantly, compared to other benchmarks based on data selection and pruning, it offers better training stability and (potentially) improved data privacy. The proposed instance-level paradigm may also inspire new approaches for data condensation in other unlabeled, low-level vision tasks. We recommend that future work should focus on further performance improvement and speeding up the condensation process.

---

[1]We cannot solely remove Group loss as the Pair loss is based on the Group loss in our method.

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

## A  APPENDIX

## B  ALGORITHM DETAILS

**Grouping and pairing.** For group assignment and pairing, we utilize the K-means algorithm (Arthur & Vassilvitskii, 2006) to obtain multiple group centroids, and compute the distance between the synthetic features and the group centroids. Specifically, after extracting and normalizing features (we chose the highest level network block's output), we compute the distance between the average of each small local feature patch (partitioned from the synthetic feature patches) and the centroids of real feature groups (obtained via K-means clustering). Each synthetic patch is then assigned to the group from which the corresponding distance (from the patch to the group centroid) is minimized, with a constraint that the number of allocated synthetic patches per group is proportional to the number of real feature patches per group. For pairing, within each group, we further compute the pairwise L2 distances between synthetic and real patches within the same group, and assign each synthetic patch to its closest real patch. We only map one real patch to at most one synthetic patch, ensuring the synthetic-to-real mapping is one-to-one.

**Warm-up and progressive assignment.** To stabilize the training process and avoid abrupt changes in the early training stage due to the grouping and pairing (which change the loss calculation), we introduce a warm-up phase, in which group and pair assignments are not performed. After that, our framework increases the number of grouped and paired synthetic patches every few hundred training iterations. At each interval, we perform the grouping step with the ungrouped features and then perform pairing with the features that were grouped in the last interval. This ensures that the features are optimized by the group loss for a period and learned toward the group feature distribution, before they are further paired with the features that are within the same group. We also add for the learned deformation, which is regularized by L2 loss, before computing the pair loss. Overall, our framework introduces different optimization objectives to different feature patches progressively, which makes the assignment process smooth and the optimization stable.

## C   IMPLEMENTATION DETAILS

### C.1   THE IMPLEMENTATION DETAILS FOR IDC FRAMEWORK

**Hyper-parameters.** By default, we partition the feature patches into smaller local feature patches with a size of 4 for grouping and pairing, and we use the last building block features for computing the distances in order to perform grouping and pairing. We use a kernel size of 7, with 256 randomly sampled channels for extracting the Random Local Fourier Features. The weights for balancing three loss terms, $w_{ins}$, $w_{group}$ and $w_{pair}$, are 300, 300 and 300k, respectively. We conducted additional experiments to validate these parameter choices, with detailed ablation studies and sensitivity analyses presented in the Subsection D.1.

**Training configurations and time.** We adopt the Adam optimizer (Maclaurin et al., 2015) in the condensation process. The learning rate is set to $1 \times 10^{-2}$, with scheduled decays at 15k, 17k, and 19k steps (milestones), where the learning rate is multiplied by a factor of $\gamma = 0.1$ at each milestone. The loss function is a weighted sum with $\alpha = 0.2$ and $\beta = 0.8$. The total training steps for condensing each class are 20k, including a 4k-step warm-up phase. In the assignment process, the number of groups is set to 16, where we increase the number of grouped and paired patches for every 400 training steps and a total of 25 times.

For the feature extractor $f$ and the upsampling ISR model $f \uparrow$, we employ SwinIR lightweight (in Table 3) (Liang et al., 2021) and MambaIRv2 (in Table 6) (Guo et al., 2024a), respectively, both of which are pre-trained on the original large-scale training dataset $\mathcal{T}$(DIV2K).

### C.2   THE IMPLEMENTATION DETAILS OF BASELINE METHODS

For all core-set selection methods, we first randomly divide the original dataset into 20 groups to improve computational efficiency. Each selection algorithm is then applied independently within each group, and the selected samples from all groups are aggregated to form the final dataset. **Random:** We randomly select a subset of samples from each group without considering any other information. This serves as a simple baseline for comparison. **Herding:** Herding (Welling, 2009) is a greedy algorithm that iteratively selects samples whose features are closest to the mean feature of the group. Specifically, at each step, the sample that, when added to the current selection, minimizes the distance between the mean of the selected features and the mean of all features in the group is chosen. **K-Center Greedy:** The K-Center Greedy (Sener & Savarese, 2018) algorithm aims to select a subset of samples such that the maximum distance from any sample in the group to its nearest selected center is minimized. We apply the K-Center Greedy algorithm: starting from a randomly chosen sample, we iteratively select the sample that is farthest from the current set of selected centers until the desired number of samples is reached.

We follow the instruction (Chengcheng Guo & Bai, 2022) to implement these methods.

**DCSR** (Ding et al., 2023) is a gradient-based dataset pruning method designed to enhance the complexity and diversity of the training set. The algorithm consists of two stages. In the first stage, the gradients of all input images are computed using the Sobel filter (Kanopoulos et al., 1988), and images with gradient values below the average are removed, thereby selecting samples with higher complexity. In the second stage, the remaining high-complexity inputs are mapped into the feature space (Zhang et al., 2017) and clustered into 15 groups. Then, 80% of the images in each group are randomly removed, resulting in only about 10% of the original data being retained for training.

It is important to note that, for all the above methods, the selection is performed on low-resolution (LR) input patches. After selection, the final training set is constructed by pairing the selected LR patches with their corresponding high-resolution (HR) patches from the original dataset $\mathcal{T}$.

### C.3   TRAINING AND TEST CONFIGURATIONS FOR ISR MODELS

To evaluate the performance of the synthetic training dataset condensed by our proposed IDC framework, we trained three representative ISR models EDSR (Lim et al., 2017), SwinIR (Liang et al., 2021) and MambaIRv2 (Guo et al., 2024a) on our synthetic dataset from scratch. The detailed architectures of these ISR models and training configurations are listed in Table 3.

After completing the training, we test the performance of these models on five test datasets under the setting of the crop_border is 2, using PSNR and SSIM as quality metrics.

Table 3: Training configurations for different ISR models in the evaluation phase.

| | network | training steps | learning rate |
|---|---|---|---|
| EDSR | upscale: 4
num_feat: 64
num_block: 16
res_scale: 1
img_range: 255.
rgb_mean: [0.4488, 0.4371, 0.4040] | 300k | 1e-4 |
| SwinIR | upscale: 4
window_size: 8
img_range: 1.
depths: [6, 6, 6, 6]
embed_dim: 60
num_heads: [6, 6, 6, 6]
mlp_ratio: 2.0
upsampler: 'pixelshuffledirect'
resi_connection: '1conv' | 500k | 2e-4 |
| Mambairv2 | upscale: 4
img_range: 1.
embed_dim: 48
d_state: 8
depths: [ 5, 5, 5, 5 ]
num_heads: [ 4, 4, 4, 4 ]
window_size: 16
inner_rank: 32
num_tokens: 64
convffn_kernel_size: 5
mlp_ratio: 1.0
upsampler: 'pixelshuffledirect'
resi_connection: '1conv' | 500k | 1e-4 |

## D    ABLATION STUDY

### D.1    ABLATION STUDY WITH THE CHOICE OF THE HYPER-PARAMETERS

Here, we provide additional ablation study results for the hyper-parameters setting, including patch size $p$, kernel size $k$ and the sampling rate used in the Random Local Fourier Features (Section 3.3). We also tested the effect of the number of layers for feature extraction and the weights for the group $w_{group}$ and pair $w_{pair}$ losses. The results are shown in the Table 4. It can be observed that the changes in PSNR and SSIM under different hyper-parameter settings are minor for different hyper parameter values. This demonstrates that our method is robust to hyper-parameter variations and can achieve stable performance.

### D.2    THE EFFECT OF THE DIFFERENT UP-SAMPLING ISR MODELS

We also evaluated the effectiveness of the different up-sampling ISR models $f \uparrow$ for the synthetic dataset. The results are shown in the Table 5 and the detailed architectures, training configurations of these ISR models are listed in Table 6. As shown in Table 5, the choice of ISR up-sampling model (EDSR, SwinIR, MambaIRv2) has only resulted in a minor impact on the final performance. For example, although MambaIRv2 is generally considered a stronger ISR model than SwinIR, we observe that many of the models trained on datasets up-sampled by SwinIR can still outperform those up-sampled by MambaIRv2. This indicates that the up-sampling ISR model used for our proposed data condensation is not sensible to the test ISR model in the evaluation phase.

Table 4: Ablation study of hyper-parameters. Each variant changes only one parameter from the default (a0).

| Variant | Patch size | Kernel size | Sampling rate | Feature layer | $w_{group}$ | $w_{pair}$ | Set5 PSNR | Set5 SSIM | Urban100 PSNR | Urban100 SSIM | Manga109 PSNR | Manga109 SSIM |
|---|---|---|---|---|---|---|---|---|---|---|---|---|
| a0 (default) | **4** | **7** | **256** | **last** | **300** | **300k** | 30.02 | 0.8616 | 24.07 | 0.7474 | 28.00 | 0.8723 |
| a1 | 2 | - | - | - | - | - | -0.18 | -0.0025 | -0.10 | -0.0042 | -0.14 | -0.0028 |
| a2 | 8 | - | - | - | - | - | -0.15 | -0.0014 | -0.02 | -0.0002 | -0.11 | -0.0014 |
| a3 | - | 5 | - | - | - | - | -0.22 | -0.0018 | -0.03 | -0.0002 | -0.06 | -0.0005 |
| a4 | - | 9 | - | - | - | - | -0.21 | -0.0028 | -0.09 | -0.0033 | -0.15 | -0.0022 |
| a5 | - | - | 128 | - | - | - | -0.14 | -0.0016 | -0.03 | -0.0012 | -0.09 | -0.0014 |
| a6 | - | - | 512 | - | - | - | -0.09 | -0.0010 | -0.03 | -0.0014 | -0.10 | -0.0015 |
| a7 | - | - | - | - | 30 | - | -0.39 | -0.0050 | -0.13 | -0.0046 | -0.22 | -0.0035 |
| a8 | - | - | - | - | 3000 | - | -0.10 | -0.0020 | -0.04 | -0.0019 | +0.08 | +0.0009 |
| a9 | - | - | - | - | - | 30k | -0.12 | -0.0018 | -0.02 | -0.0019 | +0.07 | +0.0006 |
| a10 | - | - | - | - | - | 3000k | -0.41 | -0.0053 | -0.13 | -0.0048 | -0.25 | -0.0038 |
| a11 | - | - | - | first | - | - | -0.35 | -0.0034 | -0.03 | -0.0007 | -0.07 | -0.0011 |

Table 5: Comparison with different the up-sampling ISR models $f \uparrow$ with condense ratio $r$=10%. The results are based on PSNR (dB) and SSIM. The best result is highlighted in red .

| PSNR (dB)↑ | Set5 (Bevilacqua et al., 2012) | | | Set14 (Zeyde et al., 2012) | | | Urban100 (Huang et al., 2015) | | |
|---|---|---|---|---|---|---|---|---|---|
| | EDSR | SwinIR | MambaIRv2 | EDSR | SwinIR | MambaIRv2 | EDSR | SwinIR | MambaIRv2 |
| **IDC (EDSR)** | 30.25 | 30.36 | 30.48 | 26.66 | 26.73 | 26.80 | 24.51 | 24.72 | 24.87 |
| **IDC (SwinIR)** | 30.25 | 30.39 | 30.56 | 26.64 | 26.83 | 26.92 | 24.49 | 24.90 | 25.04 |
| **IDC (MambaIRv2)** | 30.18 | 30.34 | 30.52 | 26.63 | 26.78 | 26.91 | 24.47 | 24.86 | 25.16 |

| PSNR (dB)↑ | BSD100 (Martin et al., 2001) | | | Manga109 (Fujimoto et al., 2016) | | |
|---|---|---|---|---|---|---|
| | EDSR | SwinIR | MambaIRv2 | EDSR | SwinIR | MambaIRv2 |
| **IDC (EDSR)** | 26.27 | 26.33 | 26.38 | 28.60 | 28.92 | 29.13 |
| **IDC (SwinIR)** | 26.26 | 26.37 | 26.42 | 28.59 | 29.10 | 29.21 |
| **IDC (MambaIRv2)** | 26.24 | 26.34 | 26.40 | 28.54 | 29.02 | 29.26 |

| SSIM↑ | Set5 (Bevilacqua et al., 2012) | | | Set14 (Zeyde et al., 2012) | | | Urban100 (Huang et al., 2015) | | |
|---|---|---|---|---|---|---|---|---|---|
| | EDSR | SwinIR | MambaIRv2 | EDSR | SwinIR | MambaIRv2 | EDSR | SwinIR | MambaIRv2 |
| **IDC (EDSR)** | 0.8658 | 0.8680 | 0.8693 | 0.7445 | 0.7478 | 0.7488 | 0.7620 | 0.7702 | 0.7747 |
| **IDC (SwinIR)** | 0.8658 | 0.8689 | 0.8707 | 0.7443 | 0.7503 | 0.7517 | 0.7621 | 0.7771 | 0.7811 |
| **IDC (MambaIRv2)** | 0.8644 | 0.8680 | 0.8702 | 0.7445 | 0.7487 | 0.7516 | 0.7610 | 0.7749 | 0.7824 |

| SSIM↑ | BSD100 (Martin et al., 2001) | | | Manga109 (Fujimoto et al., 2016) | | |
|---|---|---|---|---|---|---|
| | EDSR | SwinIR | MambaIRv2 | EDSR | SwinIR | MambaIRv2 |
| **IDC (EDSR)** | 0.7148 | 0.7175 | 0.7184 | 0.8805 | 0.8861 | 0.8877 |
| **IDC (SwinIR)** | 0.7147 | 0.7202 | 0.7212 | 0.8804 | 0.8892 | 0.8893 |
| **IDC (MambaIRv2)** | 0.7137 | 0.7184 | 0.7198 | 0.8795 | 0.8882 | 0.8895 |

Although we chose MambaIRv2 for our main experiments for $f \uparrow$, this was primarily to ensure compatibility and flexibility with a wide range of ISR models. Our method is designed to easily adapt to future advances in ISR architectures. The most computationally expensive part is the one-off condensation of LR patches, which does not involve the teacher model. If a superior SR model emerges in the future, we simply perform a quick inference to generate an updated set of HR patches—typically taking only a few hours—which can effectively raise the performance ceiling with minimal effort.

## D.3 ABLATION STUDY OF RLFF VS. ALTERNATIVES

We conducted an ablation study to validate our novel Random Local Fourier Features (RLFF) against alternative transformations. Our framework requires transformations that can preserve spatial structures while extracting local information, i.e., the outputs of the transformation such be feature maps. We compared RLFF against three variants: (x1) identity function without RLFF, (x2) DCT applied to convolutional filters (The standard DCT does not preserve the spatial structures in its output), and (x3) DWT with spatial resolution preservation. The results are presented in the Table 7 and demonstrate that RLFF consistently outperforms all alternatives across different datasets, validating our design for effective local feature distribution modeling.

Table 6: Training configurations for different up-sampling ISR models $f \uparrow$ for HR up-sampling.

| | network | training steps | learning rate |
|---|---|---|---|
| EDSR | upscale: 4
num_feat: 256
num_block: 32
res_scale: 0.1
img_range: 255.
rgb_mean: [0.4488, 0.4371, 0.4040] | 300k | 1e-4 |
| SwinIR | upscale: 4
window_size: 8
img_range: 1.
depths: [6, 6, 6, 6, 6, 6]
embed_dim: 180
num_heads: [6, 6, 6, 6, 6, 6]
mlp_ratio: 2.0
upsampler: 'pixelshuffle'
resi_connection: '1conv' | 500k | 2e-4 |
| MambaIRv2 | upscale: 4
img_range: 1.
embed_dim: 174
d_state: 16
depths: [6, 6, 6, 6, 6, 6, 6, 6, 6]
num_heads: [6, 6, 6, 6, 6, 6, 6, 6, 6]
window_size: 16
inner_rank: 64
num_tokens: 128
convffn_kernel_size: 5
mlp_ratio: 2.0
upsampler: 'pixelshuffle'
resi_connection: '1conv' | 500k | 1e-4 |

Table 7: Performance comparison of different feature transformation methods. Values shown are differences from the RLFF baseline.

| variant | Set5 | | Urban100 | | Manga109 | |
|---|---|---|---|---|---|---|
| | PSNR | SSIM | PSNR | SSIM | PSNR | SSIM |
| RLFF (default) | 30.02 | 0.8616 | 24.07 | 0.7474 | 28.00 | 0.8723 |
| x1 (w/o RLFF) | -0.32 | -0.0059 | -0.23 | -0.0110 | -0.17 | -0.0044 |
| x2 (DCT filters) | -0.10 | -0.0017 | -0.05 | -0.0036 | -0.02 | -0.0009 |
| x3 (Wavelets) | -0.58 | -0.0107 | -0.64 | -0.0228 | -0.52 | -0.0095 |

## E  ADDITIONAL RESULTS

### E.1  ADDITIONAL RESULTS WITH 1% TRAINING DATA

We also provide the results on the SwinIR with the condense ratio=1% on DIV2K compared with Random and DCSR (Ding et al., 2023). As shown in Table 8, our proposed IDC method achieves the best performance among all methods at both condense ratios. Notably, when the condense ratio decreases from 10% to 1%, the performance drop of IDC is the smallest compared to Random and DCSR. This demonstrates that our method is more robust and data efficient, and can better preserve performance with a high condensation ratio.

### E.2  ADDITIONAL RESULTS ON OTHER DATASET

We also conducted an additional experiment on the Flickr2K (Lim et al., 2017) dataset (2650 images in total), and provided the results in Table 9 with a 1% condensation ratio. The results further confirm our findings on DIV2K - our IDC method significantly outperforms baselines like Uniform Selection

Table 8: Comparison with coreset selection and dataset pruning methods with condense ratio $r$=1%. For our IDC method, we generate synthetic dataset for all classes with the condense ratio $r$=1%. The results are based on PSNR (dB) and SSIM.

| | condense ratio | Set5 | | Set14 | | Urban100 | | BSD100 | | Manga109 | |
|---|---|---|---|---|---|---|---|---|---|---|---|
| | | PSNR | SSIM | PSNR | SSIM | PSNR | SSIM | PSNR | SSIM | PSNR | SSIM |
| Whole | - | 30.28 | 0.8679 | 26.78 | 0.7492 | 24.92 | 0.7769 | 26.34 | 0.7185 | 28.98 | 0.8877 |
| Random | 10% | 30.20 | 0.8664 | 26.68 | 0.7469 | 24.70 | 0.7707 | 26.28 | 0.7172 | 28.74 | 0.8854 |
| | 1% | 29.68 | 0.8594 | 26.038 | 0.7330 | 23.74 | 0.7423 | 25.80 | 0.7042 | 27.65 | 0.8715 |
| DCSR | 10% | 30.33 | 0.8679 | 26.68 | 0.7478 | 24.78 | 0.7737 | 26.31 | 0.7178 | 28.87 | 0.8866 |
| | 1% | 29.90 | 0.8622 | 26.22 | 0.7362 | 23.96 | 0.7484 | 25.96 | 0.7074 | 27.83 | 0.8738 |
| **IDC (ours)** | 10% | 30.34 | 0.8680 | 26.78 | 0.7487 | 24.86 | 0.7749 | 26.34 | 0.7184 | 29.02 | 0.8882 |
| | 1% | 30.18 | 0.8658 | 26.64 | 0.7461 | 24.48 | 0.7623 | 26.25 | 0.7152 | 28.57 | 0.8827 |

Table 9: Additional results on Flickr2K (Lim et al., 2017) dataset with condense ratio $r$=1% and evaluated on SwinIR model. The results are based on PSNR (dB) and SSIM.

| | condense ratio | Set5 | | Set14 | | Urban100 | | BSD100 | | Manga109 | |
|---|---|---|---|---|---|---|---|---|---|---|---|
| | | PSNR | SSIM | PSNR | SSIM | PSNR | SSIM | PSNR | SSIM | PSNR | SSIM |
| Whole(Flickr2K) | - | 30.39 | 0.8688 | 26.85 | 0.7505 | 24.97 | 0.7779 | 26.36 | 0.7193 | 29.13 | 0.8891 |
| Random | 1% | 30.19 | 0.8659 | 26.57 | 0.7444 | 24.45 | 0.7634 | 26.22 | 0.7151 | 28.53 | 0.8823 |
| DCSR | 1% | 30.24 | 0.8662 | 26.63 | 0.7453 | 24.49 | 0.7652 | 26.24 | 0.7160 | 28.60 | 0.8831 |
| **IDC (ours)** | 1% | 30.34 | 0.8678 | 26.75 | 0.7482 | 24.84 | 0.7738 | 26.35 | 0.7183 | 29.01 | 0.8878 |

and DCSR. This demonstrates that the effectiveness of our framework is not limited to a single dataset.

## F    QUALITATIVE COMPARISONS

**Additional samples generated from the IDC framework.** Here, we provide additional samples generated from our proposed IDC framework in Figure 5, which shows multiple generated LR and HR pairs from original images. As shown in these examples, our framework is capable of generating high fidelity samples that contain the visual features similar to those in the original image, and for producing per-instance sample patches with high diversity. These characteristics allow the resulting synthetic dataset to closely mimic the original dataset feature distribution, contributing to the final high performance of the trained ISR models.

**Additional samples between different variants of the IDC framework.** We also provide additional generated samples which are obtained from different variants of our framework, used in the ablation study in Figure 6. These samples confirm that all of our contributions summarized in Section 1 provide substantial visual quality improvement, and they all contribute to the final performance as evaluated in Section 4.

**Comparison between ISR models trained with datasets from different methods.** We provide the visual comparison between the ISR models trained with the datasets, generated/sampled from different methods Figure 7, Figure 8 and Figure 9. The results here support our quantitative results, where the synthetic dataset generated by our IDC framework can provide a consistent performance improvement for all tested ISR models.

## G    LIMITATION

**Condensation efficiency**. Currently, condensing a single class (image) requires 20,000 steps, which takes about 2 GPU hours on an NVIDIA A40. To condense the entire DIV2K dataset, the total computational cost is approximately 1,600 GPU hours. Although this is just a one-off process, and the condensed dataset is fixed for ISR model training, the initial time investment remains substantial. Therefore, further optimization of the condensation process is necessary to improve efficiency.

**Upper bound of dataset performance.** In our current approach, we generate LR patches and then use a pre-trained ISR model to up-sample them into HR patches. As shown in Table 5, the choice

of ISR upsampling model has a limited impact on the dataset's performance. We note that this is also an inherent limitation for applying distribution matching to the super-resolution task, where direct condensation of high-resolution images is not feasible due to the large amount of resources required. Our method, which utilizes an ISR model to efficiently obtain HR images and effectively performs knowledge distillation (Hinton et al., 2015), is able to closely match the performance of models trained on the condensed dataset to those trained on the original dataset. However, if more advanced ISR models are developed in the future, it would be necessary to re-perform the upsampling process from LR synthetic patches to their HR versions. However, we don't need to redo the most time-consuming condensation operation.

**Data privacy.** While IDC has the potential to enhance data privacy by generating synthetic datasets that do not directly expose original sensitive data, as illustrated in Figure 5, this aspect is not the main focus of our current work. We acknowledge that privacy-preserving properties of synthetic data are an important direction, especially for applications involving personal or confidential information. However, a thorough investigation of privacy guarantees, potential risks of data leakage, and the effectiveness of IDC in various privacy-sensitive scenarios is beyond the scope of this paper, and we leave a more comprehensive study of data privacy aspects, including formal privacy analysis and empirical evaluation, for future work (Dong et al., 2022).

## H  BROADER IMPACT

Our data condensation method, designed for generating smaller synthetic datasets, can reduce the need for using large-scale real databases for algorithm training. This helps lower the carbon footprint and saves storage resources in the development phase. In practical applications like video streaming, our approach can also potentially improve the perceptual quality of streamed content if super-resolution techniques are involved, leading to a better user experience.

## I  DATASET/CODE LICENSE.

We list the licenses of all datasets and the code used in this paper in Table 10.

Table 10: The links and licenses of all datasets and the used in our paper.

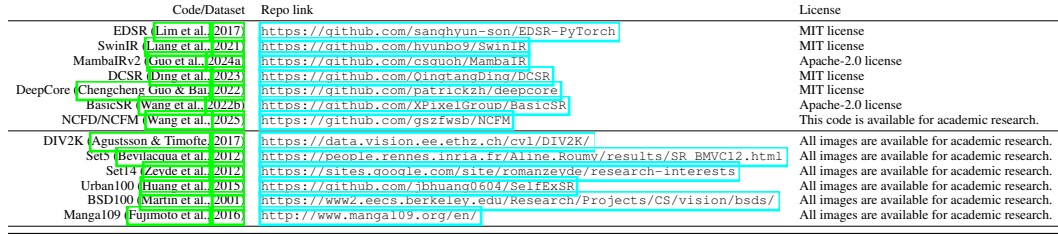

| Code/Dataset | Repo link | License |
|---|---|---|
| EDSR (Lim et al., 2017) | https://github.com/sanghyun-son/EDSR-PyTorch | MIT license |
| SwinIR (Liang et al., 2021) | https://github.com/hyunbo9/SwinIR | MIT license |
| MambaIRv2 (Guo et al., 2024a) | https://github.com/csguoh/MambaIR | Apache-2.0 license |
| DCSR (Ding et al., 2023) | https://github.com/QingtangDing/DCSR | MIT license |
| DeepCore (Chengcheng Guo & Bai, 2022) | https://github.com/patrickzh/deepcore | MIT license |
| BasicSR (Wang et al., 2022b) | https://github.com/XPixelGroup/BasicSR | Apache-2.0 license |
| NCFD/NCFM (Wang et al., 2025) | https://github.com/gszfwsb/NCFM | This code is available for academic research. |
| DIV2K (Agustsson & Timofte, 2017) | https://data.vision.ee.ethz.ch/cvl/DIV2K/ | All images are available for academic research. |
| Set5 (Bevilacqua et al., 2012) | https://people.rennes.inria.fr/Aline.Roumy/results/SR_BMVC12.html | All images are available for academic research. |
| Set14 (Zeyde et al., 2012) | https://sites.google.com/site/romanzeyde/research-interests | All images are available for academic research. |
| Urban100 (Huang et al., 2015) | https://github.com/jbhuang0604/SelfExSR | All images are available for academic research. |
| BSD100 (Martin et al., 2001) | https://www2.eecs.berkeley.edu/Research/Projects/CS/vision/bsds/ | All images are available for academic research. |
| Manga109 (Fujimoto et al., 2016) | http://www.manga109.org/en/ | All images are available for academic research. |

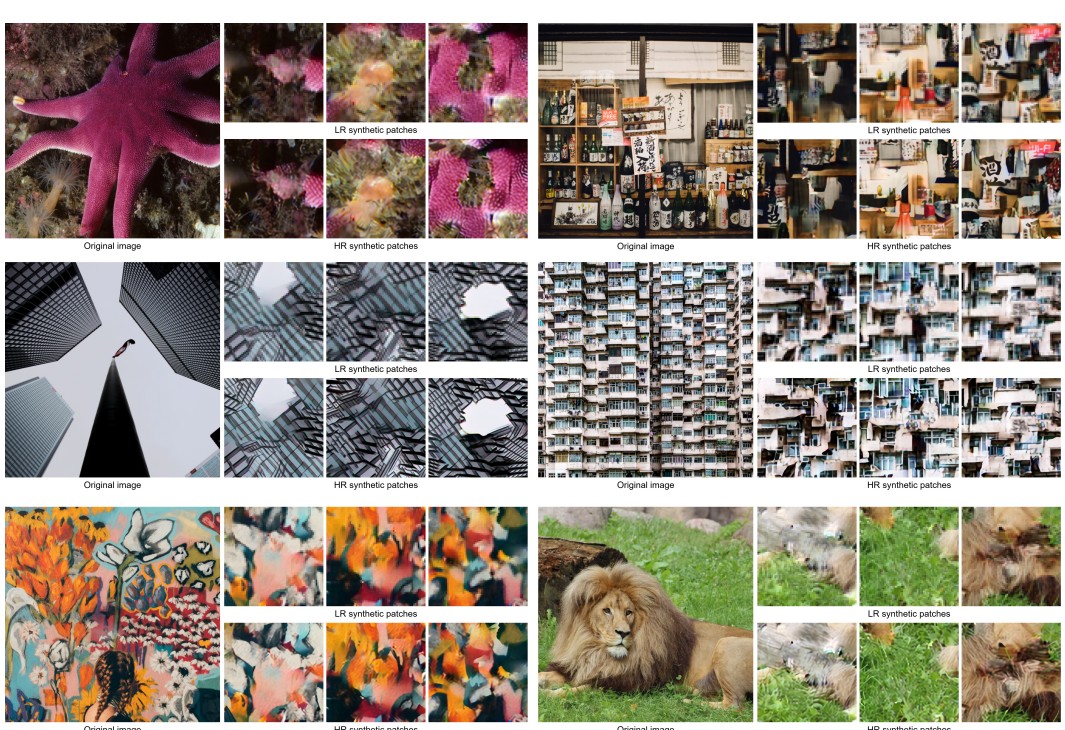

Figure 5: Additional visual results of the proposed IDC framework.

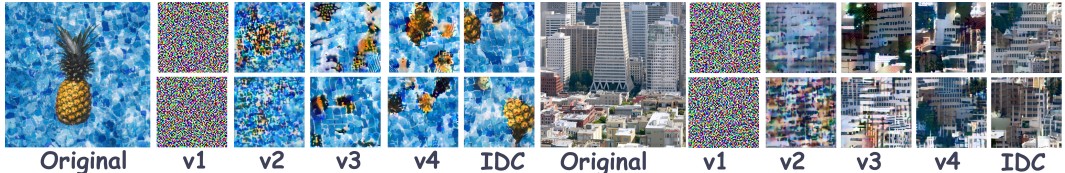

Figure 6: Additional qualitative comparison between different variants.

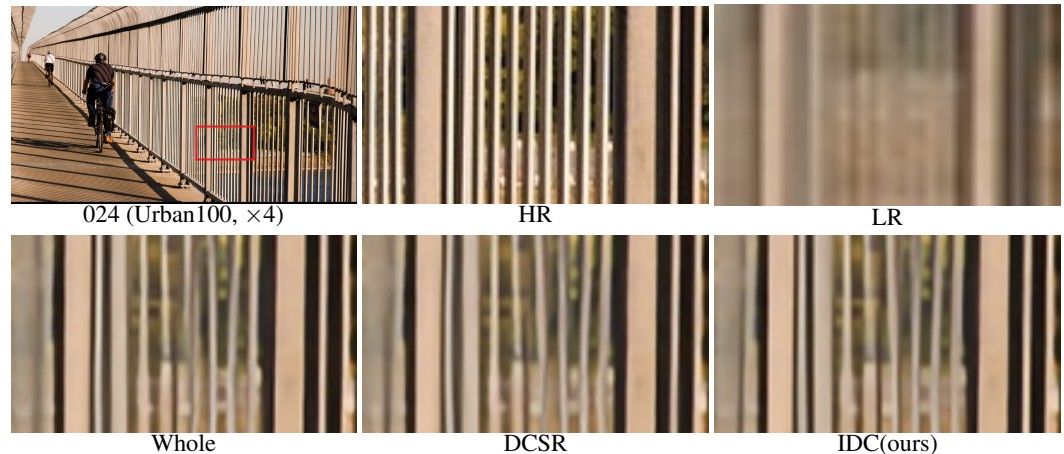

Figure 7: Qualitative comparison between different methods evaluated on EDSR (Lim et al., 2017)

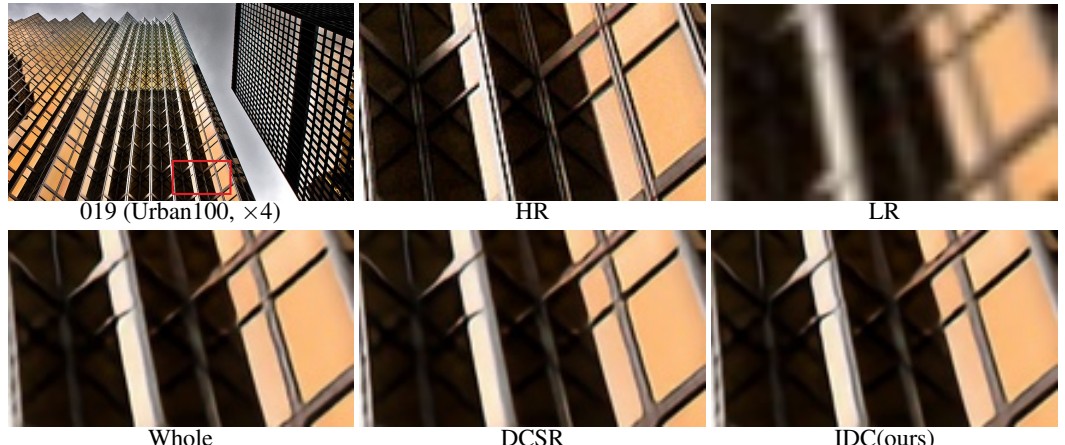

Figure 8: Qualitative comparison between different methods evaluated on SwinIR (Liang et al., 2021)

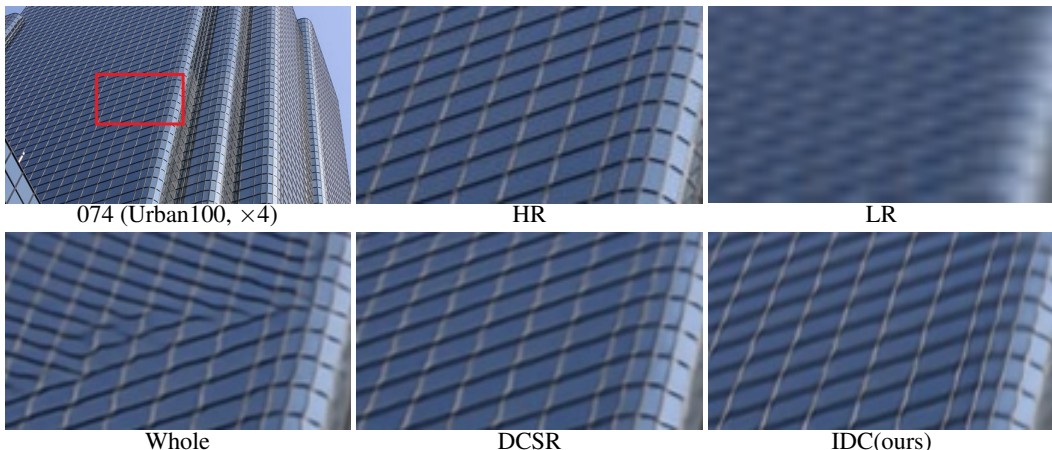

Figure 9: Qualitative comparison between different methods evaluated on MambaIRv2 (Guo et al., 2024a)

# J QUANTITATIVE ANALYSIS

## J.1 ANALYSIS OF GENERALIZATION PERFORMANCE

We further evaluated the generalization performance of the SR model (SwinIR (Liang et al., 2021)) trained on the condensed dataset produced by our framework. Figure 10 and Figure 11 show the training and validation curves in terms of L1 loss and PSNR. The training curves are computed on the condensed DIV2K (Timofte et al., 2017) dataset, and the validation curves are computed on the five validation datasets used in the main paper. In all cases, the training and validation curves are well aligned: both improve as training progresses and we do not observe any degradation over time. This indicates that no overfitting occurs and suggests that the model trained on our compact condensed dataset generalizes well, and continues to learn features that are helpful for the task, rather than simply memorizing the condensed set.

Note that the training and validation curves lie in different loss/PSNR ranges. This can be caused by several factors: (i) our condensed dataset does not contain real ground-truth high resolution images, and the learning target are generated by the teacher model, and (ii) the condensed dataset contains synthesized content instead of only real images. As a result, the absolute metric values on the condensed dataset are not directly comparable to those on the real validation benchmarks.

## J.2 ANALYSIS OF TRAINING EFFICIENCY

To quantitatively evaluate the training efficiency of our proposed framework, we compared the validation trajectories of a SwinIR model (Liang et al., 2021) trained on our 10% condensed dataset (IDC), the full dataset ("Whole"), a 10% DCSR subset (Ding et al., 2023), and a 10% uniformly sampled subset. As illustrated by the PSNR curves in Figure 12, our IDC-trained model demonstrates significantly faster convergence. To quantify this, Table 11 shows that our method consistently reaches target PSNR milestones in as little as 25% to 50% of the training steps required by the full dataset. Crucially, our method not only learns faster but also achieves a final performance that is superior or highly comparable to training on the complete dataset, while consistently outperforming the other 10% methods. This provides strong evidence that our IDC method produces a highly efficient and information-dense dataset, accelerating training without sacrificing—and in many cases, even improving—final model performance.

Table 11: Comparison of training iterations (in thousands, 'k') required to reach target PSNR values. The target PSNR values are selected base on the full dataset training.

| Test Dataset | Target PSNR | Whole | Uniform (10%) | DCSR (10%) | IDC (10%) |
|---|---|---|---|---|---|
| Set5 | 29.38 | 20k | 20k | 15k | **10k** |
| | 29.99 | 60k | 60k | 35k | **20k** |
| Set14 | 26.00 | 15k | 15k | 15k | **10k** |
| | 26.54 | 50k | 55k | 40k | **20k** |
| Urban100 | 24.18 | 45k | 45k | 30k | **20k** |
| | 24.67 | 135k | 180k | 105k | **55k** |
| BSD100 | 25.55 | 10k | 10k | 10k | **5k** |
| | 26.07 | 35k | 35k | 25k | **15k** |
| Manga109 | 28.11 | 45k | 50k | 30k | **15k** |
| | 28.69 | 125k | 150k | 90k | **40k** |

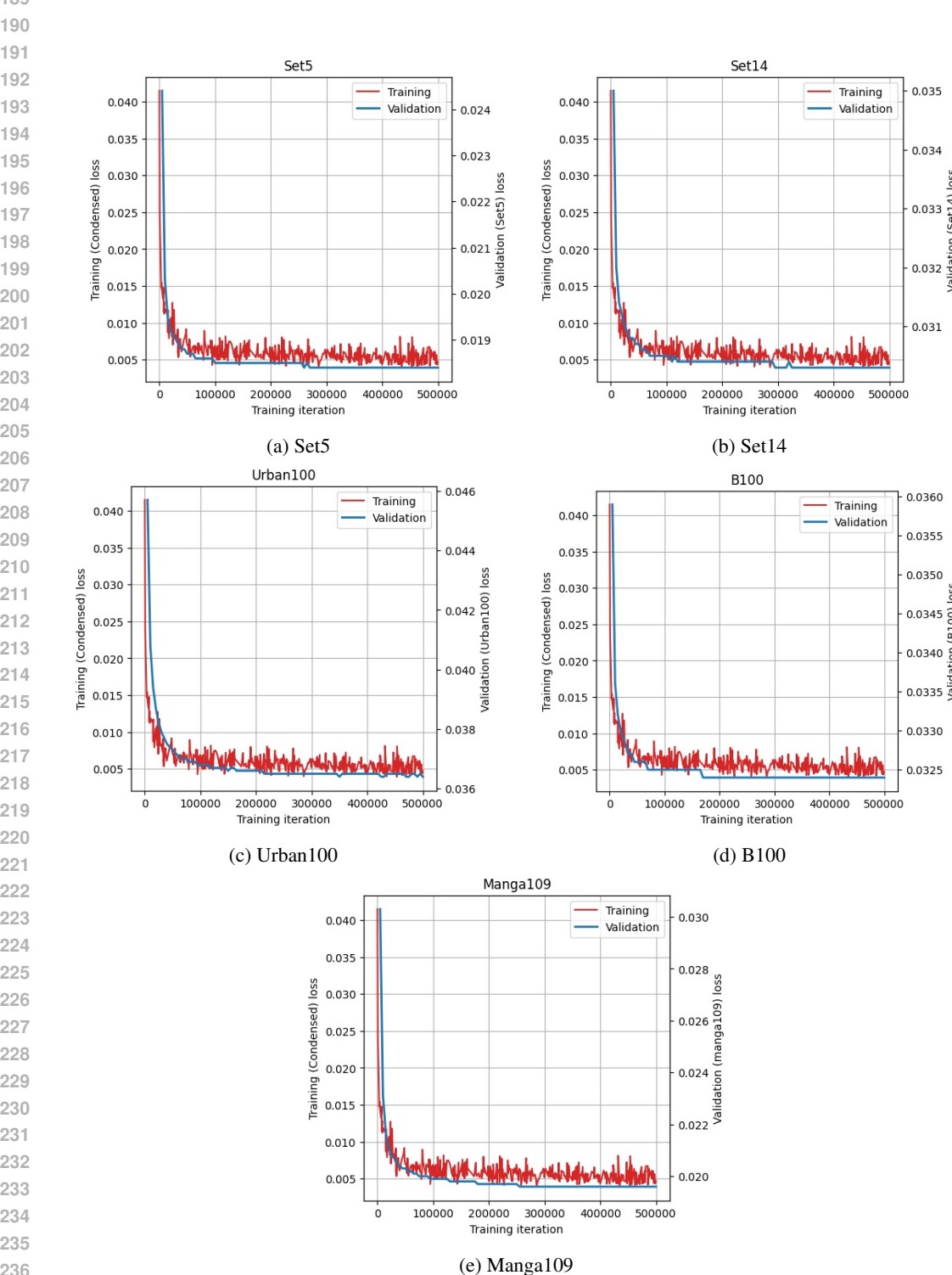

(a) Set5

(b) Set14

(c) Urban100

(d) B100

(e) Manga109

Figure 10: Train and validation loss of the model (SwinIR) trained with our condensed dataset. Validation are performed across five datasets.

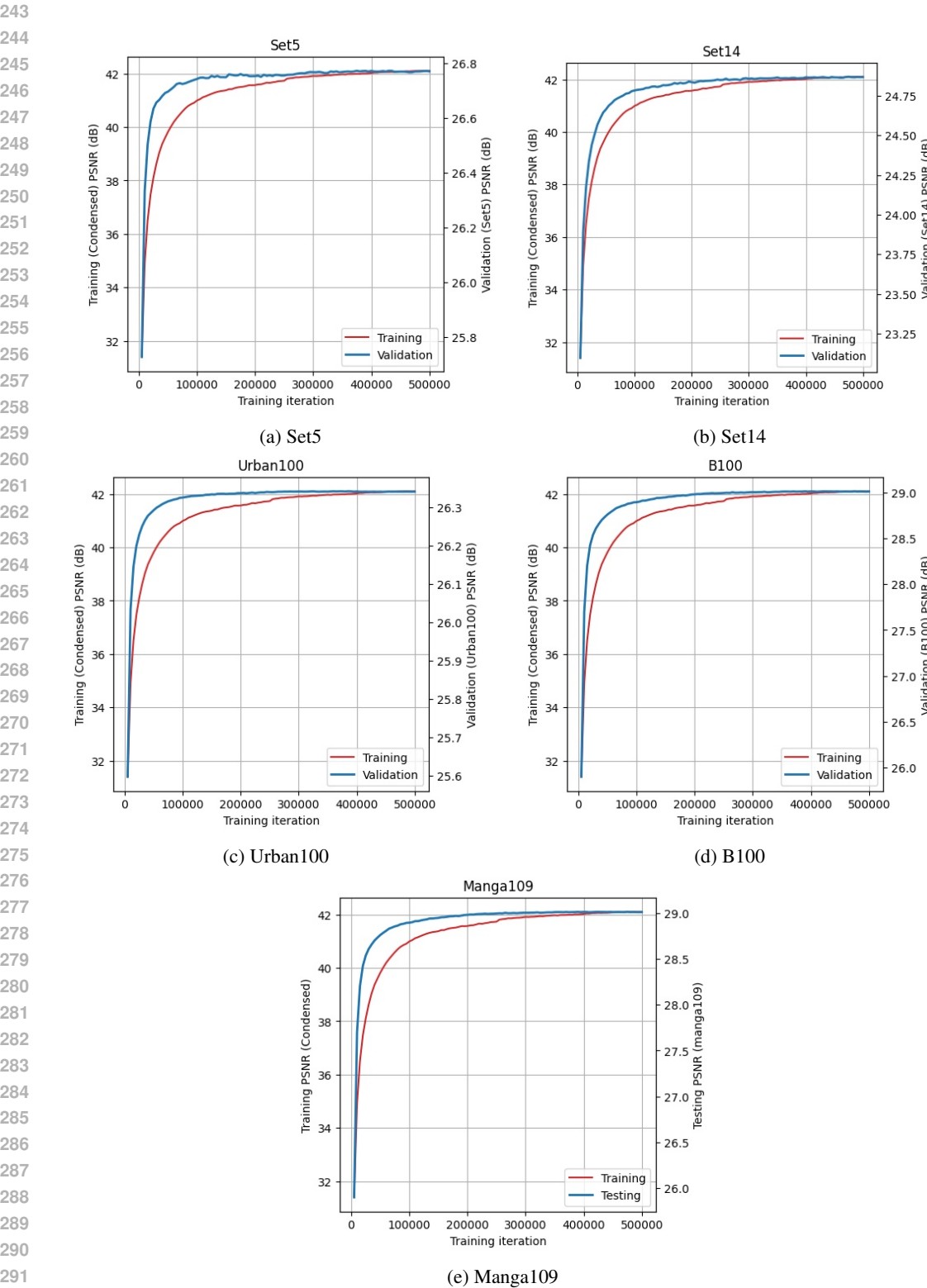

(a) Set5

(b) Set14

(c) Urban100

(d) B100

(e) Manga109

Figure 11: Train and validation PSNR of the model (SwinIR) trained with our condensed dataset. Validation are performed across five datasets.

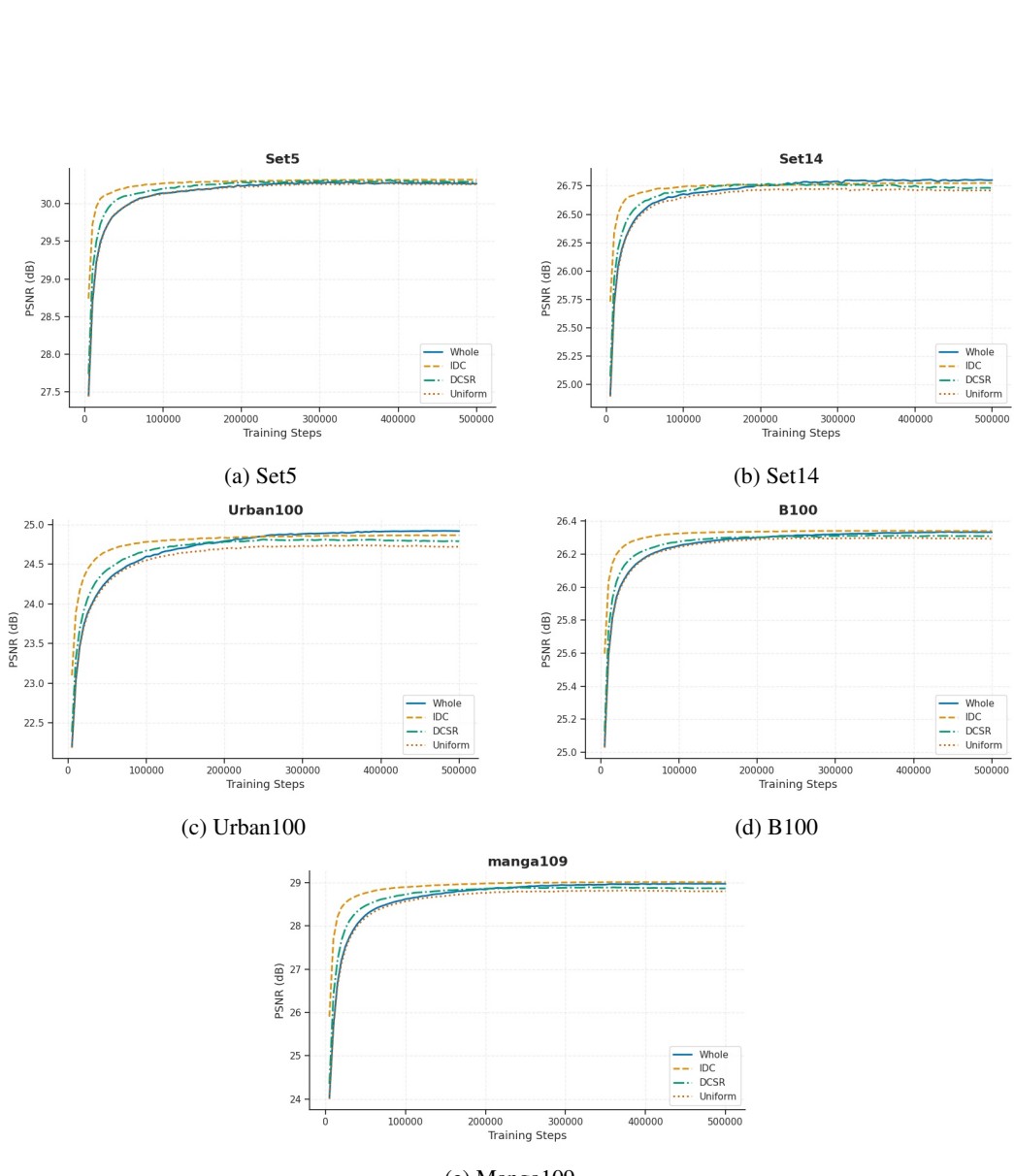

(a) Set5

(b) Set14

(c) Urban100

(d) B100

(e) Manga109

Figure 12: Validation trajectory for different methods (training datasets) across five datasets.

