# OpenReview forum: "Instance Data Condensation for Image Super Resolution"
_ICLR.cc/2026/Conference — Submitted to ICLR 2026_

### Official Review · Reviewer_AS8f · 2025-11-01

**Soundness:** 3
**Presentation:** 1
**Contribution:** 3
**Rating:** 4
**Confidence:** 4

**Summary:**

This paper introduces Instance Data Condensation (IDC), a novel framework tailored for image super-resolution (ISR). The method introduces two main components: Random Local Fourier Feature Extraction, which preserves high-frequency local details crucial for ISR, and Multi-level Feature Distribution Matching, which aligns feature distributions at both instance and group levels to maintain diversity and fidelity in the synthesized data. Experiments conducted on the DIV2K dataset with a 10% condensation rate show that the synthetic dataset achieves comparable or even superior performance to the full dataset when training state-of-the-art ISR models such as EDSR, SwinIR, and MambaIRv2. The work represents the first successful application of dataset condensation to low-level vision, demonstrating efficient data compression without loss of model quality or training stability.

**Strengths:**

This paper proposes a new data condensation framework, Instance Data Condensation, to the best of my knowledge, is the first to apply the concept of dataset condensation to image super-resolution (ISR). While numerous condensation methods have been explored in high-level vision tasks such as classification, detection, and segmentation, similar attempts have not been made for low-level vision problems. Therefore, the proposed method shows a degree of novelty and exploratory value in extending data condensation to the SR domain.

**Weaknesses:**

1. The paper’s presentation is quite poor, making it difficult to follow the main ideas. The Related Work and Methodology sections are intermixed, with prior studies and the proposed approach discussed together without clear separation. This confuses the reader and obscures the novelty of the work.
2. Beyond presentation issues, the paper also suffers from conceptual ambiguity in several analyses. For example, in the discussion of Figure 1 (left), the authors claim that the DCSR method suffers from a bias toward complex textures, yet the selected “snow mountain” region actually corresponds to a structured area rather than a purely textured one. Moreover, the figure does not convincingly show that the proposed IDC method avoids such bias. Similarly, the claim that transforming features into the Fourier domain leads “to a more informative representation” (line 259) is unsubstantiated and conceptually weak—Fourier transformation changes the representation domain but does not inherently increase information content. These unclear or overstated interpretations undermine the analytical rigor of the paper and should be supported by clearer quantitative evidence or theoretical reasoning.
3. Another concern arises from the ablation study (Table 2). The results for variants V5–V7 show larger performance drops compared with V4, even though each variant removes different components of the proposed framework (e.g., Unfolding, Local Feature, or Instance/Group Losses). This trend appears inconsistent with the claim that these components are beneficial, since removing them does not lead to clearly distinguishable or interpretable degradations.

**Questions:**

The paper claims that using a 10% condensed dataset significantly improves training efficiency. However, no quantitative evidence is provided. Could the authors clarify how the training time, number of iterations, and computational cost compare between training on the condensed dataset and the full (“Whole”) dataset?

---

> ### Author Response · Authors · 2025-11-24
>
> **(W1) Paper presentation.**
>
> Sorry for the confusion caused by our example and description.
>
> Our original intention for the structure of the Methodology section was to first establish the unique challenges of dataset condensation for ISR (i.e., the optimization difficulties associated with high-resolution patches and the absence of class labels). This was meant to logically lead into why we chose a distribution matching approach, then to introduce the baseline metric (NCFD), analyze its limitations for ISR (e.g., global information fusion and inability to capture high-frequency details), and finally, to present our novel components as direct solutions to these identified problems.
>
> However, we understand that this narrative structure inadvertently blurred the lines between prior work and our contributions, obscuring the paper's novelty. **In the revised manuscript, we will restructure these sections.
>
> **Specifically, we will move the preliminary discussions into a seperated section**, which includes the technical details of the prior methods that closely related to our work. This will allow the **Methodology section to focus exclusively and clearly on detailing our proposed framework and its novel components**. We hope this will improve the presentation of the paper and make it easy to follow.
>
> **(W2) Figure examples and theoretical analysis of RLFF.**
>
> **Regarding the "bias" figure example**, the use of the term "bias toward complex textures" is based on the official implementation and claims of the DCSR method itself [Ding et al., 2023]. As detailed in our Appendix (Subsection C.2), DCSR is a two-stage pruning method: it first preserves patches with high texture complexity (using a Sobel filter) and then further clusters and prunes them. The authors themselves claim this process remove patches with "low or similar texture complexity". While the "snow mountain" example is indeed a structured area, it is also a region of high texture complexity, which is why it was selected by their algorithm. A key distinction is that DCSR **extracts real patches**, which will naturally contain structure, whereas our method **synthesizes patches from scratch**.
>
> Your observation that Figure 1 does not visually prove that IDC avoids such bias is correct. The stronger, quantitative evidence lies in our training stability analysis. As shown in the **validation trajectory in Figure 3 (right)** (line443), the model trained on the DCSR dataset begins to overfit very early. We hypothesize that this is due to the dataset's bias towards only high-complexity content, which hinders the model's generalization. In contrast, our IDC-trained model learns the diverse feature distribution with distribution matching, and shows a stable and continuous learning trend, indicating a more balanced and generalizable dataset.
>
> In the revised version, we will rephrase this discussion for clarity and strive to present more diverse and convincing visual examples to better illustrate this point.
>
> **For the theoretical analysis of RLFF**, the combination of RLFF and the unfolding operation is designed to address the high-frequency texture modeling problem. In the original NCFD [Wang et al., 2025], images or patches (e.g. $N \times H \times W \times C_1$) are treated as high dimensional samples (i.e., $N$ samples each with $H \times  W \times C_1$ dimensions), resulting in an intractable distribution ($\mathbb{R}^{H \times  W \times  C_1}$) due to the high sample dimensionality, which also makes the learning of the high-frequency details particularly difficult. Here, we resolve this problem by modeling the local feature distributions: RLFF extracts local features in different frequencies $(N \times H \times W \times C_1 → N \times H \times W \times C_2)$; the unfolding operation, i.e., the step that unfolds spatial features to the batch dimension ($(N \times H \times W \times C_2)$ → ($N \times H/P \times W/P)$ samples with size $P \times P \times C_2$, $P$ is the patch size), treats each point of the transformed features as individual samples in a batch. As a result, our framework models the local feature distributions ($\mathbb{R}^{P \times  P \times  C_2}$), instead of those of the full image/patch ($\mathbb{R}^{H \times  W \times  C_1}$), which is much more efficient.
>
> Moreover, the superiority of RLFF for this task is quantitatively validated in Appendix D.3, where RLFF consistently outperforms other alternatives, such as DCT and DWT.

---

> > ### Author Response · Authors · 2025-11-24
> >
> > **(W3) Ablation study overlapping.**
> >
> > Thank you for this very insightful question regarding our ablation study. Your observation that the performance drops for variants V5-V7 is significant. We appreciate the opportunity to explain this.
> >
> > In our design, there are different components providing various features, but their contributions could overlap, which causes what you observe in the ablation study; i.e., removing RLFF (V5) or removing unfolding (V6) does not cause a significant performance drop. RLFF and unfolding operations are designed to address different issues, both of which are rooted by the nature of the task—learning high resolution images with fine details.
> >
> > - **Purpose of RLFF (V5 - No Unfolding, but with RLFF):** RLFF was specifically designed to replace the random Gaussian mapping used in the original NCFD. As we state in the paper (line 215), this is because NCFD's global projection struggles to capture the high-frequency features essential for ISR. RLFF transforms features locally where high-frequency content can be extracted easier, which is why simply adding RLFF to the baseline (from v1) yields a significant performance boost.
> >
> > - **Purpose of Unfolding (V6 - No RLFF, but with Unfolding):** The Unfolding operation was designed to make the distribution matching problem itself tractable. As noted (line 210), learning the global feature distribution of a high-resolution image $(N \times C \times H \times W)$ is intractable. Unfolding reshapes this into a batch of local patches $((N \times H/P \times W/P) \times C \times P \times P)$, allowing our loss functions to operate on manageable, local distributions instead of an intractable global one.
> >
> > To address this comment, we have performed an **additional experiment** by removing both RLFF and unfolding, while keeping the other components (i.e., the multi-level matching losses). In this case, the framework is not able to learn any meaningful condensation output, i.e., the SR model trained with the condensation output performance is around 10 dB on all test datasets (as the synthetic patches are almost pure noise which is similar to the v1 on Figure 4 (line 456) ).
> >
> > We hope this detailed explanation clarifies the ablation results. Thank you again for pushing us to provide this deeper analysis.

---

> > > ### Author Response · Authors · 2025-11-24
> > >
> > > **(Q1) Quantitative results of training efficiency.**
> > >
> > > Thank you for pointing out that our initial claim about "training efficiency" was not sufficiently supported by quantitative evidence. We appreciate the opportunity to provide a detailed comparison.
> > >
> > > When we refer to "significant improvements in training efficiency," our primary claim is centered on **faster convergence**—that is, the ability of a model to reach a target performance level in a fraction of the training steps required when using the full dataset. This allows for quicker iterations and more efficient monitoring of a model's performance during development.
> > >
> > > To provide the quantitative evidence, we provided the table that compares the number of training iterations (in thousands, 'k') required for a SwinIR model to reach specific target PSNR values on five standard test datasets These PSNR milestone values are set by the model trained with the full and non-condensed dataset. We compare our 10% IDC dataset against the full ("Whole") dataset and other 10% selection-based methods.
> > >
> > > | Test Dataset | Target PSNR | Whole | Uniform (10%) | DCSR (10%) | **IDC (10%)** |
> > > | :--- | :--- | :--- | :--- | :--- | :--- |
> > > | **Set5** | 29.38 | 20k | 20k | 15k | **10k** |
> > > | | 29.99 | 60k | 60k | 35k | **20k** |
> > > | **Set14** | 26.00 | 15k | 15k | 15k | **10k** |
> > > | | 26.54 | 50k | 55k | 40k | **20k** |
> > > | **Urban100** | 24.18 | 45k | 45k | 30k | **20k** |
> > > | | 24.67 | 135k | 180k | 105k | **55k** |
> > > | **BSD100** | 25.55 | 10k | 10k | 10k | **5k** |
> > > | | 26.07 | 35k | 35k | 25k | **15k** |
> > > | **Manga109** | 28.11 | 45k | 50k | 30k | **15k** |
> > > | | 28.69 | 125k | 150k | 90k | **40k** |
> > >
> > > Across all datasets and at various performance thresholds, the model trained on our 10% IDC dataset consistently reaches the target PSNR with **significantly fewer iterations**—often requiring only **25% to 50%** of the training steps compared to the model trained on the full dataset. Note that the computational cost per iteration is the same across all settings. Thus, our method effectively improves training efficiency.
> > >
> > > To further illustrate this, we included the **complete validation trajectories for all five test datasets** in the Figure 12 of the Appendix (line 1306) of our revised manuscript.
> > >
> > > Regarding the **total training time and computational cost** for the main experiments in our paper, we followed the standard practice[Cazenavette et al., 2022, Wang et al., 2025] for a fair comparison by training all models for the **same total number of iterations** (e.g., 500k for SwinIR). The detailed configurations are provided in Appendix Table 3 (line 759). Our efficiency claim is not about reducing the total training time in *that specific experimental setup*, but about the **rate of learning**. The faster convergence demonstrated above means that for practical applications, one could achieve a desired performance level much more quickly.
> > >
> > > We hope this detailed explanation and the new quantitative data fully resolve your concern. We apologize again for the initial ambiguity and will be much more precise in our revised paper.
> > >
> > > **Reference**
> > >
> > > [Cazenavette et al., 2022] Cazenavette, George, et al. "Dataset distillation by matching training trajectories." Proceedings of the IEEE/CVF Conference on Computer Vision and Pattern Recognition. 2022.
> > >
> > > [Ding et al., 2023] Ding, Qingtang, et al. "Not all patches are equal: Hierarchical dataset condensation for single image super-resolution." IEEE Signal Processing Letters 30 (2023): 1752-1756.
> > >
> > > [Wang et al., 2025] Wang, Shaobo, et al. "Dataset distillation with neural characteristic function: A minmax perspective." Proceedings of the Computer Vision and Pattern Recognition Conference. 2025.

---

### Official Review · Reviewer_vLAv · 2025-11-01

**Soundness:** 3
**Presentation:** 3
**Contribution:** 3
**Rating:** 6
**Confidence:** 3

**Summary:**

This paper introduces a novel framework termed Instance Data Condensation (IDC) for image super-resolution. IDC addresses the challenge of reducing training data volume while maintaining or even enhancing model performance. The framework leverages Random Local Fourier Features (RLFF) and Multilevel Feature Distribution Matching to condense training datasets at the instance level, eliminating the need for class labels common in high-level vision tasks. Extensive experiments and ablation studies validate the effectiveness and robustness of the proposed method.

**Strengths:**

1. This paper is well-motivated and easy to follow.
2. The proposed framework achieves better performance with only 10% synthetic crops.

**Weaknesses:**

1. The condensation process is computationally intensive.
2. Although the instance-level paradigm is promising, its effectiveness across diverse tasks remains to be validated.
3. The scalability of the IDC framework across datasets of different volumes lacks empirical validation.

**Questions:**

1. Does the IDC data distillation method affect the generalization performance of super-resolution models? Please provide relevant experimental results to illustrate.
2. What is the memory footprint of RLFF?
3. When the condensation ratio falls below 10%, is any modal collapse observed? Or is the model overfitting?

---

> ### Author Response · Authors · 2025-11-24
>
> **(W1) Cost of Condensation.**
>
> Thank you for raising this important point. The computational cost of the condensation process is indeed significant, which we have already discussed in the **Limitations** section of our Supplementary. Here, we would like to emphasize several key points in this context:
>
> 1. It is important to note that the condensation process is a **one-time, upfront investment** to generate a highly reusable dataset.
>
> 2. High computational cost is a common characteristic of existing dataset condensation methods. Our IDC method takes approximately 1.5 GPU-hours per class/instance on an NVIDIA 4090 GPU to generate 15 images per class (IPC=15). This cost is highly competitive when compared to other high-fidelity methods in high-level vision. For instance, the influential DC method [Kim et al., 2022] requires approximately 2 hours per class to condense the CIFAR-10 dataset (IPC=10) on a single RTX-2080 GPU. Furthermore, even methods designed for scalability, such as TESLA [Cui et al., 2023], can be very computationally intensive, requiring approximately 10.5 GPU hours per class to condense the CIFAR-100 dataset on a single A6000 GPU.
>
> Moreover, the high cost of our method are also attributed to four primary factors inherent to low level vision tasks such as ISR:
>
> - **High Resolution with fine details:** Low-level vision tasks operate on higher resolution images compared to the smaller images (e.g., $32 \times32$) that are often used in classification-focused condensation research. Moreover, our generated images contain fine details that are close to the original dataset, where high-level vision data condensation methods typically generate images that contain more semantic information but coarser details. This inherently increases the computational load.
>
> - **Powerful Feature Extractor:** To capture the fine-grained details necessary for ISR, we use a powerful pre-trained model (SwinIR) as a feature extractor. This model, with its millions of parameters, is vastly more complex than small lightweight models(e.g., the simple ConvNet) that are often used in other condensation studies
> [Su et al., 2024, Liu et al., 2025, Wang et al., 2025].
>
> - **Synthesis from Scratch:** Our method synthesizes new data points from scratch to optimize the information content. This is fundamentally more demanding than simpler coreset selection or pruning methods.
>
> - **Targeting performance for practical use:** The condensed dataset generated by our method can offer performance comparable to that of the standard dataset when used to train SoTA SR models,  while existing condensation methods cannot provide such performance.
>
> 3. More importantly, **the value of condensation** extends beyond the final performance. It unlocks several potential benefits:
>
>     - **Improved Training Efficiency & Faster Convergence:** A primary benefit is the ability to reach or even exceed the performance of the full dataset in a fraction of the training time. To provide concrete evidence, the table below shows the validation trajectory on the challenging Manga109 test set.
>     |Dataset/Steps|50K|100K|150K|200K|250K|300K|350K|400K|450K|500K|
>     |---|---|---|---|---|---|---|---|---|---|---|
>     |IDC (10%)|28.57|28.81|28.89|28.94|28.97|28.98|28.99|28.99|29.00|29.00|
>     |Full DIV2K|28.20|28.61|28.75|28.85|28.91|28.94|28.96|28.97|28.98|28.98|
>
>    The data clearly shows that model trained on our IDC condensed dataset **converges significantly faster and achieves a higher final PSNR**. We have included the full validation trajectories across all five test datasets in the revised Appendix (Figure 12 (line 1306)).
>
>     - **Enhanced Model Generalization & Stability:** Our condensed dataset promotes better training behavior. As shown in Figure 3 (line 443) in our paper, our method avoids the overfitting in selection-based methods at a 1% ratio.
>
>     - **Reduced storage:** Fundamentally, a 90% reduction in dataset size directly translates to significantly lower storage costs and data transfer overhead.
>
> 4. **Future Directions for Efficiency:** We acknowledge that improving condensation efficiency is a crucial and exciting direction for future research. One promising avenue could be **training-free alternatives**. For instance, one might consider leveraging powerful **generative models** such as **diffusion models**. However, directly adapting these models to generate synthetic images that are not only high-quality and rich in detail but also faithfully **condense** the information from a specific source dataset remains a non-trivial challenge. Nevertheless, exploring how to harness such powerful generative priors for efficient one-shot dataset condensation presents a highly promising avenue for future work.
>
> In summary, while the initial cost is high, we believe it is a meaningful trade-off for the substantial and lasting benefits in training efficiency and storage that our condensed dataset provides.

---

> > ### Author Response · Authors · 2025-11-24
> >
> > **(W2/W3) More large-scale datasets and other low-level vision tasks.**
> >
> > Thank you for this valuable suggestion. We agree that the benefits of dataset condensation are more evident when testing on large-scale datasets.
> >
> > The core principle of our method—instance-level condensation from unlabeled data—is inherently generic. For tasks like image deblurring and image denoising, this is particularly relevant, as they often rely on large training datasets without class labels, as in ISR. Given the time and resource constraints during the rebuttal period, we prioritized validating our method's extensibility on one of these tasks. We conducted an new experiment on **color image denoising** on a larger dataset**. The promising results from that experiment serve as a strong proof-of-concept for our framework's applicability to other low-level vision tasks.
> >
> > Following the common practice for image denoising, as in SwinIR [Liang et al., 2021], we chose a larger training set containing **DIV2K (800), Flickr2K (2650), BSD500 (400), and WED (4744 images)**, with a total of **8,594 images**. Due to resource and time constraints during the rebuttal, we performed a condensation with a **1% ratio** using our IDC framework, and trained a SwinIR model on this 1% condensed dataset to compare its performance against the models trained on the full dataset, a uniform selected 10% subset, and a uniform selected 1% subset.
> >
> > Specifically, to provide a more technical discussion on how we implemented this extension for image denoising, here are the specific details of our scheme:
> >
> > - **Synthesizing the Noisy Input**: The noisy, low-quality (LQ) patches for a given noise level $\sigma$ are treated as learnable synthetic parameters. The feature extractor \(f\) is a powerful pre-trained denoising model (we used a SwinIR model trained on the original dataset for denoising [Liang et al., 2021]).
> >
> > - **Distribution Matching**: Our IDC framework optimizes these synthetic LQ patches by matching their feature distributions to those of the real LQ patches. The "real LQ" set is created by taking the clean, high-quality (HQ) images from the original dataset and applying the corresponding fixed noise level $\sigma$.
> >
> > - **Generating the $HQ_{syn}$**: Note that the $LQ_{syn}$ patches learned are already noisy; i.e., our method implicitly learns the image degradation model. The corresponding $HQ_{syn}$ patches for our synthetic set are generated by passing the optimized synthetic $LQ_{syn}$ through the same teacher denoiser model so we do not apply noise manually to the condensed images when training the corresponding denoising model.
> >
> > - **Knowledge Distillation**: This end-to-end process ensures that the resulting $(LQ_{syn}, HQ_{syn})$ pairs are not just simple training data, but implicitly encode the complex mapping and feature statistics inherent learned in the teacher model, effectively distilling the teacher's knowledge into the condensed set.
> >
> > Adapting our framework to the denoising task also requires a few practical modifications due to the different task requirements:
> >
> > - **Larger Patch Size:** Denoising models are typically trained on larger patches (eg. $128 \times 128$).
> >
> > - **Larger Feature Extractor:** The commonly used SOTA models for denoising (e.g., the full SwinIR, not a lightweight version) are significantly larger, leading to a substantial increase in feature dimensionality and memory consumption during condensation.
> >
> > - **Our Adjustment:** To manage this computational burden and perform the experiments during the rebuttal period,  we have adjusted the condensation settings, such as reducing the number of iterations (10k steps) and adjusting the hyper-parameter of RLFF (i.e., sampling rate, $256 \rightarrow128$) to speed up the condensation process.
> >
> > The preliminary results are as follows:
> >
> > | Method | McMaster | Kodak24 | CBSD68 | BSD68 | Urban100 |
> > |--------|----------|---------|--------|-------|----------|
> > | | **PSNR/SSIM** | **PSNR/SSIM** | **PSNR/SSIM** | **PSNR/SSIM** | **PSNR/SSIM** |
> > | Whole | 35.53 dB; 0.9335 | 35.28 dB; 0.9292 | 34.37 dB; 0.9351 | 34.83 dB; 0.9399 | 35.04 dB; 0.9516 |
> > | Uniform(10%) | 35.48 dB; 0.9328 | 35.26 dB; 0.9288 | 34.35 dB; 0.9347 | 34.83 dB; 0.9399 | 34.99 dB; 0.9510 |
> > | Uniform(1%) | 35.47 dB; 0.9327 | 35.24 dB; 0.9287 | 34.31 dB; 0.9341 | 34.78 dB; 0.9392 | 34.95 dB; 0.9507 |
> > | IDC(1%) | 35.51 dB; 0.9316 | 35.26 dB; 0.9291 | 34.36 dB; 0.9348 | 34.83 dB; 0.9399 | 35.00 dB; 0.9506 |
> >
> > As the results indicate, the SwinIR model trained on our **1% condensed dataset consistently outperforms the one trained on a 1% uniformly sampled subset and, remarkably, achieving similar or even better performance to the 10% subset.**
> >
> > This experiment strongly suggests that the benefits of our IDC framework can indeed scale to larger datasets and extend to other low-level vision tasks. We appreciate the suggestion of the reviewer, which enabled us to explore this promising direction.

---

> ### Author Response · Authors · 2025-11-24
>
> **(Q1) The effect of proposed IDC method on the generalization performance of ISR models.**
>
> Thank you for your insightful question. The generalization is, indeed, an important indicator of performance. We found that our model trained with the condensed dataset generalizes well to the evalutaiton datasets.
>
> In the revised Appendix, we provide the plots (Figures 10 and 11) of the training and validation performance (measured in loss/PSNR) over training iterations. We find that the **training and validation performance are highly correlated, and both of them improve over the whole training process**. There are also **no signs of overfitting**. This indicates that the ISR model generalizes to the evaluation datasets.
>
> Moreover, in Appendix Table 8, we show that our proposed IDC has better performance than the baseline methods even with a 1% condensation ratio. It is worth noting that the same Table 8 also shows that the absolute performance drop from condensation ratio 10% to 1% is smaller with IDC (e.g., **24.86dB → 24.48dB (-0.38dB) for IDC**, whereas **24.70dB → 23.74dB (-0.96dB) and 24.78dB → 23.96dB (-0.82dB)** are observed for random selection and DCSR, when measuring in PSNR and with Urban100). The better performance at lower condensation ratios indicates that IDC-condensed datasets are helpful for generalization, so that the performance of the trained models is much better even with a very small number of training samples.
>
>
> **(Q2) Memory footprint of RLFF.**
>
> RLFF is, in fact, memory efficient. It is implemented as a single convolution operation (the Fourier transform, sampling, and convolution are linear and can be fused). It requires a total of additional **~2.2 GB** memory (with FP32) in our standard setting and implementation, which involves feature matching at multiple layers. Note that it is relatively small compared with the memory cost of the feature extractor and the other operations (e.g., **~8.4GB** when with SwinIR), as the feature extractor contains many layers (24 blocks, each with multiple convolution/MLP/attention layers), and the activations are required to be cached for optimizing the synthesized images.

---

> > ### Author Response · Authors · 2025-11-24
> >
> > **(Q3) Overfitting with smaller condensation ratio.**
> >
> > We have investigated the performance and training stability of our method at a more aggressive **1% condensation ratio**. Our findings, which we have included in the paper, demonstrate the robustness of our approach.
> >
> > As shown in **Table 8** (line 921) and **Figure 3** (line 443) in our paper, we compared the performance of models trained on datasets generated by our IDC method, Random Selection, and DCSR, all at a 1% ratio. The validation trajectory on the Set14 dataset (Figure 3, right) is particularly revealing.
> >
> > To better illustrate this, the table below shows the full validation trajectory on Set14 at the challenging 1% condensation ratio.
> >
> > |Dataset/Steps|50K|100K|150K|200K|250K|300K|350K|400K|450K|500K|
> > |---|---|---|---|---|---|---|---|---|---|---|
> > |Uniform (1%)|26.42|26.42|26.37|26.35|26.33|26.25|26.20|26.19|26.16|26.16|
> > |DCSR (1%)|26.48|26.46|26.44|26.41|26.35|26.30|26.27|26.25|26.23|26.22|
> > |IDC (1%)|26.55|26.46|26.66|26.67|26.67|26.67|26.67|26.69|26.69|26.68|
> >
> > While selection-based methods (Uniform, DCSR) exhibit clear overfitting (performance degrades over time), our **IDC method maintains a stable and upward learning trend**, demonstrating superior generalization even in an extremely limited data.
> >
> > **Reference**
> >
> > [Liang et al., 2021] Liang, Jingyun, et al. "Swinir: Image restoration using swin transformer." Proceedings of the IEEE/CVF international conference on computer vision. 2021.
> >
> > [Kim et al., 2022] Kim, Jang-Hyun, et al. "Dataset condensation via efficient synthetic-data parameterization." International Conference on Machine Learning. PMLR, 2022.
> >
> > [Cui et al., 2023] Cui, Justin, et al. "Scaling up dataset distillation to imagenet-1k with constant memory." International Conference on Machine Learning. PMLR, 2023.
> >
> > [Su et al., 2024] Su, Duo, et al. "D^ 4: Dataset Distillation via Disentangled Diffusion Model." Proceedings of the IEEE/CVF Conference on Computer Vision and Pattern Recognition. 2024.
> >
> > [Liu et al., 2025] Liu, Ping, and Jiawei Du. "The evolution of dataset distillation: Toward scalable and generalizable solutions." arXiv preprint arXiv:2502.05673 (2025).
> >
> > [Wang et al., 2025] Wang, Shaobo, et al. "Dataset distillation with neural characteristic function: A minmax perspective." Proceedings of the Computer Vision and Pattern Recognition Conference. 2025.

---

### Official Review · Reviewer_p8PL · 2025-11-03

**Soundness:** 3
**Presentation:** 3
**Contribution:** 2
**Rating:** 4
**Confidence:** 3

**Summary:**

This paper proposes a new dataset condensation framework for image super-resolution, by designing a Multi-level Feature Distribution Matching approach and Random Local Fourier Features. The conducted experiments show that the condensed datasets give promising performance.

**Strengths:**

1. The motivation is clear. This paper shows a new dataset condensation method for image super-resolution.
2. The writing of this paper is fluent, and the content is easy to follow.
3. The designs of the two approaches make sense to some degree.
4. The conducted ablation experiments are detailed and well-designed.

**Weaknesses:**

1. More large-scale datasets should be condensed to show the promising performance of the proposed method. The related datasets in the paper are DIV2K and Flickr2K, which are not very large in real scenarios. I believe that the value of condensation is more evident on large-scale datasets than in experiments with specific case studies.
2. The performance improvement is not very obvious, as shown in Table 1, and the condensation burden comparison should be provided to give more analysis.
3. Can you give some theoretical analysis or insights about your designs, such as random local Fourier features?
4. Can this method extend to other low-level missions, such as deblur and denoise? Please give some discussions.

**Questions:**

Please refer to "Weaknesses".

---

> ### Author Response · Authors · 2025-11-24
>
> **(W1) More large-scale datasets and other low-level vision tasks.**
>
> Thank you for this valuable suggestion. We agree that the benefits of dataset condensation are more evident when testing on large-scale datasets.
>
> Similar to high-level vision tasks, successfully condensing very large datasets remains a significant challenge for low-level vision.  For instance, even state-of-the-art methods face a considerable performance gap on classification tasks [Liu et al., 2025]. On **Tiny ImageNet**, a recent SOTA method [Su et al., 2024] using a ResNet-18 backbone with a 10% condensation ratio (IPC=50) achieves an accuracy of **46.2%**, which is substantially lower than the that (**61.9%**) when training on the full dataset. Similarly, on the much larger **ImageNet-1K** dataset, a condensed set with a 10% ratio (IPC=100) yields **59.3%** accuracy with the same backbone, compared to **69.8%** on the whole dataset. These examples shows that effective large-scale condensation without a major drop in performance, is still a highly challenging research frontier.
>
> In the context of image super-resolution, datasets like DIV2K (800 images) and Flickr2K (2650 images) are widely considered as large-scale training sets. Nevertheless, inspired by your valuable suggestion (also linking to Weakness 4 - application to other low level tasks),  to further explore the potential of our method, we conducted a new experiment on a **larger**, combined dataset for the task of **color image denoising**.
>
> Following the common practice for image denoising, as in SwinIR [Liang et al., 2021], we chose a larger training set containing **DIV2K (800), Flickr2K (2650), BSD500 (400), and WED (4744 images)**, with a total of **8,594 images**. Due to resource and time constraints during the rebuttal, we performed a condensation with a **1% ratio** using our IDC framework, and trained a SwinIR model on this 1% condensed dataset to compare its performance against the models trained on the full dataset, a uniform selected 10% subset, and a uniform selected 1% subset.
>
> The preliminary results are as follows:
>
> | Method | McMaster | Kodak24 | CBSD68 | BSD68 | Urban100 |
> |--------|----------|---------|--------|-------|----------|
> | | **PSNR/SSIM** | **PSNR/SSIM** | **PSNR/SSIM** | **PSNR/SSIM** | **PSNR/SSIM** |
> | Whole | 35.53 dB; 0.9335 | 35.28 dB; 0.9292 | 34.37 dB; 0.9351 | 34.83 dB; 0.9399 | 35.04 dB; 0.9516 |
> | Uniform(10%)  | 35.48 dB; 0.9328 | 35.26 dB; 0.9288 | 34.35 dB; 0.9347 | 34.83 dB; 0.9399 | 34.99 dB; 0.9510 |
> | Uniform(1%)  | 35.47 dB; 0.9327 | 35.24 dB; 0.9287 | 34.31 dB; 0.9341 | 34.78 dB; 0.9392 | 34.95 dB; 0.9507 |
> | **IDC(1%)**| **35.51 dB; 0.9316** | **35.26 dB; 0.9291** |**34.36 dB; 0.9348**| **34.83 dB; 0.9399**| **35.00 dB; 0.9506** |
>
>
> As the results indicate, the SwinIR model trained on our **1% condensed dataset consistently outperforms the one trained on a 1% uniformly sampled subset and, remarkably, achieving similar or even better performance to the 10% subset.** This demonstrates the superior data efficiency of our method. While a performance gap to the full dataset still remains, such competitive results (with only 1/100 of the data) have already shown the great promise of the proposed method.
>
> This experiment strongly suggests that the benefits of our IDC framework can indeed scale to larger datasets and extend to other low-level vision tasks. We appreciate the suggestion of the reviewer, which enabled us to explore this promising direction.

---

> > ### Author Response · Authors · 2025-11-24
> >
> > **(W2) Performance improvement and condensation burden.**
> >
> > We acknowledge that the performance improvements in Table 1 may appear subtle at first glance, and we would like to provide more context regarding both the significance of these gains and the value proposition of the condensation burden.
> >
> > 1. **Context of Performance Gains in ISR:** The field of image super-resolution is characterized by small but incremental gains, where improvements of around 0.1 dB are often considered noteworthy. To our knowledge, our method is the first to demonstrate that a model trained on a dataset comprising only **10% of the original volume** can achieve performance that is not only comparable but sometimes superior to that trained on the full dataset. This achievement is particularly notable when contrasted with the current state of dataset condensation in high-level vision. As we mentioned in our response to **W1**, even SOTA methods there struggle to close the performance gap. For example, on ImageNet-1K, a 10% condensed set (IPC=100) still results in a >10% absolute drop in accuracy (59.3% vs. 69.8%) for a ResNet-18, where ResNet-18 is not even a common baseline for the image classification task in the standard (non-condensation) setting. The fact that our method closes and even surpasses this gap in the ISR task with modern models (e.g. SwinIR) highlights its effectiveness.
> >
> > 2. **Context of computation burden in main condensation research**: Our IDC method takes approximately 1.5 GPU-hours per class/instance on an RTX4090 GPU to generate 15 images per class (IPC=15). This cost is highly competitive when compared to other high-fidelity methods in high-level vision. For instance, the influential DC method [Kim et al., 2022] requires approximately 2 hours per class to condense the CIFAR10 dataset (IPC=10) on a single RTX2080 GPU. Furthermore, even methods designed for scalability, like TESLA [Cui et al., 2023] can be very computationally intensive, requiring approximately 10.5 GPU-hours per class to condense the CIFAR100 dataset on a single A6000 GPU. Crucially, these high-level methods typically operate on low-resolution images (e.g., 32x32 for CIFARs), while our ISR task typically requires processing higher-resolution inputs (with finer details) using a powerful feature extractor -  this increased complexity naturally makes our process more demanding.
> > Moreover, regarding performance, the condensed database generated by our method can offer comparable performance to the standard dataset when training SoTA SR models, while other condensation methods cannot.
> >
> > 3. **The value of condensation** extends beyond the final performance. It unlocks several potential benefits:
> >
> >     - **Improved Training Efficiency & Faster Convergence:** A primary benefit is the ability to reach or even exceed the performance of the full dataset in a fraction of the training time. To provide concrete evidence, the table below shows the full validation trajectory on the challenging Manga109 test set.
> >
> >     |Dataset/Steps|50K|100K|150K|200K|250K|300K|350K|400K|450K|500K|
> >     |---|---|---|---|---|---|---|---|---|---|---|
> >     |Full DIV2K|28.20|28.61|28.75|28.85|28.91|28.94|28.96|28.97|28.98|28.98|
> >     |IDC (10%)|28.57|28.81|28.89|28.94|28.97|28.98|28.99|28.99|29.00|29.00|
> >
> >    The data clearly shows that model trained on our IDC condensed dataset **converges significantly faster and achieves a higher final PSNR**. We also included the full validation trajectories across all five test datasets in the revised Appendix (Figure 12 (line 1306)).
> >
> >     - **Enhanced Model Generalization & Stability:** Our condensed dataset promotes better training behavior. As shown in Figure 3 (line 443) in our paper, our method avoids the overfitting in selection-based methods at a 1% ratio. To  better illustrate this, the table below shows the full validation trajectory on Set14 at the challenging 1% condensation ratio.
> >
> >     |Dataset/Steps|50K|100K|150K|200K|250K|300K|350K|400K|450K|500K|
> >     |---|---|---|---|---|---|---|---|---|---|---|
> >     |Uniform (1%)|26.42|26.42|26.37|26.35|26.33|26.25|26.20|26.19|26.16|26.16|
> >     |DCSR (1%)|26.48|26.46|26.44|26.41|26.35|26.30|26.27|26.25|26.23|26.22|
> >     |IDC (1%)|26.55|26.46|26.66|26.67|26.67|26.67|26.67|26.69|26.69|26.68|
> >
> >     While selection-based methods (Uniform, DCSR) exhibit clear overfitting (performance degrades over time), our **IDC method maintains a stable and upward learning trend**, demonstrating superior generalization even in an extreme low condensation ratio.
> >
> >     - **Reduced storage:** Fundamentally, a 90% reduction in dataset size directly translates to significantly lower storage costs and data transfer overhead.
> >
> >     In summary, the condensation burden is a **one-time investment** that yields a highly efficient, robust, and high-performing dataset. The benefits in terms of **faster training**, **better generalization**, and **reduced resource requirements** strongly justify the initial computational cost.

---

> > > ### Author Response · Authors · 2025-11-24
> > >
> > > **(W3) Theoretical analysis or insights about RLFF.**
> > >
> > > The combination of RLFF and the unfolding operation is designed to address the high-frequency texture modeling problem. In the original NCFD [Wang et al., 2025], images or patches (e.g. $N \times H \times W \times C_1$) are treated as high dimensional samples (i.e., $N$ samples each with $H \times  W \times C_1$ dimensions), resulting in an intractable distribution ($\mathbb{R}^{H \times  W \times  C_1}$) due to the high sample dimensionality, which also makes the learning of the high-frequency details particularly difficult. Here, we resolve this problem by modeling the local feature distributions: RLFF extracts local features in different frequencies $(N \times H \times W \times C_1 → N \times H \times W \times C_2)$; the unfolding operation, i.e., the step that unfolds spatial features to the batch dimension ($(N \times H \times W \times C_2)$ → ($N \times H/P \times W/P)$ samples with size $P \times P \times C_2$, $P$ is the patch size), treats each point of the transformed features as individual samples in a batch. As a result, our framework models the local feature distributions ($\mathbb{R}^{P \times  P \times  C_2}$), instead of those of the full image/patch ($\mathbb{R}^{H \times  W \times  C_1}$), which is much more efficient.
> > >
> > > Moreover, the superiority of RLFF for this task is quantitatively validated in Appendix D.3, where RLFF consistently outperforms other alternatives, such as DCT and DWT.

---

> > > > ### Author Response · Authors · 2025-11-24
> > > >
> > > > **(W4) Extension to other low-level tasks.**
> > > >
> > > > This is a good point, and we strongly believe that extending our Instance Data Condensation (IDC) framework to other low-level vision tasks is a promising future direction.
> > > >
> > > > The core principle of our method—instance-level condensation from unlabeled data—is inherently generic. For tasks like image deblurring and image denoising, this is particularly relevant, as they often rely on large training datasets without class labels, as in ISR. Given the time and resource constraints during the rebuttal period, we prioritized validating our method's extensibility on one of these tasks. As detailed in our response to Weakness 1, **we have conducted an experiment on color image denoising**. The promising results from that experiment serve as a strong proof-of-concept for our framework's applicability to other low-level vision tasks.
> > > >
> > > > To provide a more technical discussion on how we implemented this extension for image denoising, here are the specific details of our scheme:
> > > >
> > > > - **Synthesizing the Noisy Input**: The noisy, low-quality (LQ) patches for a given noise level $\sigma$ are treated as learnable synthetic parameters. The feature extractor \(f\) is a powerful pre-trained denoising model (we used a SwinIR model trained on the original dataset for denoising [Liang et al., 2021]).
> > > >
> > > > - **Distribution Matching**: Our IDC framework optimizes these synthetic LQ patches by matching their feature distributions to those of the real LQ patches. The "real LQ" set is created by taking the clean, high-quality (HQ) images from the original dataset and applying the corresponding fixed noise level $\sigma$.
> > > >
> > > > - **Generating the $HQ_{syn}$**: Note that the $LQ_{syn}$ patches learned are already noisy; i.e., our method implicitly learns the image degradation model. The corresponding $HQ_{syn}$ patches for our synthetic set are generated by passing the optimized synthetic $LQ_{syn}$ through the same teacher denoiser model so we do not apply noise manually to the condensed images when training the corresponding denoising model.
> > > >
> > > > - **Knowledge Distillation**: This end-to-end process ensures that the resulting $(LQ_{syn}, HQ_{syn})$ pairs are not just simple training data, but implicitly encode the complex mapping and feature statistics inherent learned in the teacher model, effectively distilling the teacher's knowledge into the condensed set.
> > > >
> > > > Adapting our framework to the denoising task also requires a few practical modifications due to the different task requirements:
> > > >
> > > > - **Larger Patch Size:** Denoising models are typically trained on larger patches (eg. $128 \times 128$).
> > > >
> > > > - **Larger Feature Extractor:** The commonly used SOTA models for denoising (e.g., the full SwinIR, not a lightweight version) are significantly larger, leading to a substantial increase in feature dimensionality and memory consumption during condensation.
> > > >
> > > > - **Our Adjustment:** To manage this computational burden and perform the experiments during the rebuttal period,  we have adjusted the condensation settings, such as reducing the number of iterations (10k steps) and adjusting the hyper-parameter of RLFF (i.e., sampling rate, $256 \rightarrow128$) to speed up the condensation process.
> > > >
> > > > The results are detailed in the response to **W1**, which clearly shows the proposed method can be extended to the denoising tasks with simple modifications. I would like to thank you once again for your thoughtful suggestion, which has certainly broadened our perspective and highlighted exciting avenues for our future work.
> > > >
> > > >
> > > > **Reference**
> > > >
> > > > [Liang et al., 2021] Liang, Jingyun, et al. "Swinir: Image restoration using swin transformer." Proceedings of the IEEE/CVF international conference on computer vision. 2021.
> > > >
> > > > [Kim et al., 2022] Kim, Jang-Hyun, et al. "Dataset condensation via efficient synthetic-data parameterization." International Conference on Machine Learning. PMLR, 2022.
> > > >
> > > > [Cui et al., 2023] Cui, Justin, et al. "Scaling up dataset distillation to imagenet-1k with constant memory." International Conference on Machine Learning. PMLR, 2023.
> > > >
> > > > [Su et al., 2024] Su, Duo, et al. "D^ 4: Dataset Distillation via Disentangled Diffusion Model." Proceedings of the IEEE/CVF Conference on Computer Vision and Pattern Recognition. 2024.
> > > >
> > > > [Liu et al., 2025] Liu, Ping, and Jiawei Du. "The evolution of dataset distillation: Toward scalable and generalizable solutions." arXiv preprint arXiv:2502.05673 (2025).
> > > >
> > > > [Wang et al., 2025] Wang, Shaobo, et al. "Dataset distillation with neural characteristic function: A minmax perspective." Proceedings of the Computer Vision and Pattern Recognition Conference. 2025.

---

### Author Response · Authors · 2025-11-30
**Global Response**

Dear Area Chair and Reviewers,

We sincerely thank all reviewers for their time and effort in evaluating our paper. We are pleased that the reviewers have acknowledged the following contributions:

*   **Strong motivation.** This work is the first to systematically apply dataset condensation to a low-level vision task - ISR, a problem acknowledged as important and well-motivated. *[Reviewers p8PL, vLAv, and AS8f]*
*   **Novel methodology.** The proposed IDC framework, including the Random Local Fourier Features (RLFF) and Multi-Level Feature Distribution Matching, has been recognized as technically sound and novel. *[Reviewers vLAv and AS8f]*
*   **Superior
 performance.** Our method is the first to demonstrate that a 10% condensed dataset can achieve performance comparable or even superior to the full dataset, a result acknowledged as promising and significant. *[Reviewers p8PL, vLAv, and AS8f]*
*   **Broad evaluation.** Our extensive experiments across multiple datasets, models, and condensation ratios were recognized as detailed and well-designed. *[Reviewers p8PL, vLAv, and AS8f]*

### **Summary of Author Responses and New Experiments**

We also thank all reviewers for their insightful and constructive suggestions, which helped us further improve the paper. The major responses and revisions are summarized below:

*   **New experiments on a large-scale (8,594 images) dataset for image denoising**, which show that our IDC framework also generalizes to larger datasets and other low-level vision tasks (denoising). Without modification to the method, our framework achieved competitive performance compared to the full set training even with just 1% ratio of the data. *[Reviewers p8PL(W1&W4), vLAv(W2&W3)]*

*   **A quantitative analysis on training efficiency**, which substantiates our claims, showing that the condensed dataset enables up to 2-4 times faster convergence to target performance (Appendix J). *[ReviewerAS8f(Q1)]*

*   **A new analysis of generalization performance** by comparing training and validation curves across five test datasets, which shows they are well-aligned and demonstrates that the model generalizes well rather than memorizing the compact set (Appendix J). *[Reviewer vLAv(Q1)]*

*   **A stability analysis at an aggressive 1% ratio**, which demonstrates that our method avoids the overfitting seen in baseline methods, showcasing its robustness in extreme low-data regimes. *[Reviewer vLAv(Q3)]*

*   **A discussion on the condensation burden** that contextualizes the one-time cost, showing its competitiveness and justifying it with substantial downstream benefits in training efficiency and generalization. *[Reviewers p8PL(W2), vLAv(W1)]*

*   **A discussion and new ablation study**, which further clarifies the distinct and crucial contributions of our core components. *[Reviewer p8PL(w3), AS8f(W2)]*

We will further revise our paper in a later revision, including the changes listed below.

*   **Including the additional experiments and analyses** provided in the author responses, e.g., the experiments on the large scale dataset and the image denoising task, and the analysis on training efficiency.

*   **Restructuring the paper's method sections** to improve clarity by separating the text regarding the preliminary discussions into a new section*[Reviewer AS8f(W1)]*

We believe these substantial additions have greatly strengthened our paper's contributions and overall rigor. Once again, we sincerely thank all the reviewers and the AC for the valuable guidance and constructive suggestions. We hope the revised manuscript and detailed responses can fully addressed all the concerns raised during the review process.

Best regards,
Authors of Paper #17902

---

### Meta-Review · Area_Chair_iu6z · 2026-01-06

**Summary:**

This paper proposes Instance Data Condensation (IDC) for SR, aiming to compress large training datasets into a smaller synthetic set while maintaining comparable performance. Motivated by the high computational and storage cost of training SR models on full datasets, the authors introduce an instance-level condensation framework that combines Random Local Fourier Feature extraction and Multi-level Feature Distribution Matching. The paper shows that a condensed dataset at a 10% ratio can achieve performance comparable to, and in some cases slightly better than, training on the full dataset, with improved training stability and faster convergence.

The initial reviewer scores are mixed, with two reviewers giving scores around the rejection threshold (4, 4) and one reviewer giving a moderately positive score (6). Reviewers’ concerns mainly focus on the scale and practical value of condensation for ISR, the computational burden of the condensation process itself, the limited theoretical insight, presentation quality, and the lack of a convincing discussion of real-world benefits.

The authors responded to the reviewers’ comments, adding new experiments on a larger low-level vision dataset (image denoising), analyses of training dynamics and convergence speed, and detailed discussions of condensation cost as a one-time upfront investment. However, none of the reviewers participated further in the discussion phase after the rebuttal.

**Reviewer Concerns:**

As the Area Chair, I read the paper, the reviewer reports, and the authors’ responses. I believe this paper needs to be interpreted from a rather specific perspective.

First, regarding the concern raised by Reviewer p8PL that the value of condensation is more evident on large-scale datasets, I largely agree with this assessment. However, for sr, the practical notion of a “large-scale” dataset is different from that in high-level vision. The datasets commonly used in SR (e.g., DIV2K, Flickr2K) are not extremely large, and SR training is not generative in the same sense as large-scale representation learning. As a result, the marginal benefit of condensation on increasingly larger datasets is difficult to evaluate and may not scale in the same way as in classification or detection. At the same time, I do not fully agree with the claim that the performance improvements reported are insignificant: in the context of SR, improvements on the order of a few hundredths of a dB can already be meaningful and worth analyzing. That said, I agree with the reviewers that the paper would benefit from deeper theoretical analysis and clearer insights into why the proposed condensation works for SR, beyond empirical demonstrations. Extending the discussion to other low-level vision tasks is also a reasonable and important direction, which the authors partially address in the rebuttal.

The most critical issue for me, however, is the question of practical significance and cost–benefit trade-offs, which is raised most clearly by Reviewer vLAv and implicitly by Reviewer AS8f. The condensation process itself is computationally expensive. While the authors argue that this cost is a one-time upfront investment, this argument depends heavily on the actual downstream training scenario. In practice, SR models are also typically trained once per configuration; training an SR model is itself already a one-time cost, and the absolute computational burden of SR training is relatively modest compared to modern large-scale generative or foundation model training.

Even after condensation, the authors’ own analysis suggests that training still requires 25%–50% of the original number of iterations. This does not change the order of magnitude of training cost. Given that SR training is not particularly “dense” or prohibitive in today’s computing environment, it remains unclear whether the proposed condensation leads to a compelling net gain in realistic settings. Moreover, the potential drawbacks of condensation—such as loss of diversity, hidden biases, or reduced flexibility for future reuse—are difficult to assess and not fully explored.

In short, while this is an interesting and intellectually appealing idea, the paper does not convincingly establish that the problem it addresses has strong and clear real-world significance. The authors should not expect reviewers or readers to infer this significance on their own; instead, the paper needs to explicitly and quantitatively justify when and why instance-level data condensation is truly beneficial for SR in practice.

Finally, I partially agree with Reviewer AS8f that the presentation of the paper requires improvement. Several sections could be structured more clearly, and some claims would benefit from more precise wording and stronger quantitative grounding, especially when discussing efficiency and practical impact.

Overall, although the paper explores a novel and intriguing direction and presents substantial experimental effort, I am not convinced that it currently demonstrates sufficiently strong practical significance or a compelling cost–benefit trade-off to warrant acceptance. For these reasons, I recommend rejection at this time.

**Reviewer Scores:**

None of the reviewers participated further in the discussion after the authors’ rebuttal. Based on the tone of the original reviews and the nature of the remaining concerns, it is unlikely that the reviewers who initially gave scores of 4 would have substantially increased their ratings. The reviewer who initially gave a score of 6 would likely have maintained a similar assessment, but there is no clear indication that they would have raised it further given the unresolved questions about practical significance and cost–benefit trade-offs.

---

### Decision · Program_Chairs · 2026-01-26

Reject